

# Measurement report: Size-Resolved and Seasonal Variations in Aerosol Hygroscopicity Dominated by Organic Formation and Aging: Insights from a Year-Long Observation in Nanjing

Junhui Zhang[1], Yuying Wang[1], Jialu Xu[1], Xiaofan Zuo[1], Chunsong Lu[1], Bin Zhu[1], Yuanjian Yang[1], Xing Yan[2], Yele Sun[3]

[1]State Key Laboratory of Climate System Prediction and Risk Management/Key Laboratory for Aerosol–Cloud Precipitation of China Meteorological Administration/Special Test Field of National Integrated Meteorological Observation, Nanjing University of Information Science & Technology, Nanjing 210044, China
[2]Faculty of Geographical Science, Beijing Normal University, Beijing 100875, China
[3]State Key Laboratory of Atmospheric Environment and Extreme Meteorology, Institute of Atmospheric Physics, Chinese Academy of Sciences, Beijing 100029, China

*Correspondence to*: Yuying Wang (yuyingwang@nuist.edu.cn)

**Abstract:** Aerosol hygroscopicity plays a significant role in atmospheric chemistry, radiation, and climate effects. While previous studies have investigated regional differences in aerosol hygroscopicity, long-term observational studies focusing on seasonal variations in specific regions remain scarce. This study explores size-resolved and seasonal variations in aerosol hygroscopicity in northern Nanjing, using one-year hygroscopicity-tandem differential mobility analyser (H-TDMA) measurements in 2021. Aerosols in the region show relatively low hygroscopicity due to a high organic mass fraction (annual average mass fraction: 42.92% in $PM_{2.5}$) in fine particles. The mean hygroscopicity parameter ($\kappa_{mean}$) increases with particle size across all seasons, with more pronounced size dependence in nucleation-mode particles. Particles (40–200 nm) show seasonal $\kappa_{mean}$ variations: winter (0.12–0.24) and spring (0.14–0.25) display higher values attributable to secondary inorganic aerosols, while summer (0.12–0.21) and autumn (0.10–0.20) exhibit weaker hygroscopicity due to enhanced contributions from less hygroscopic components. Diurnal patterns are shaped by photochemical aging and aqueous-phase reactions, leading to $\kappa_{mean}$ peaks for larger particles in the afternoon and evening. New particle formation events are most frequent in spring, producing initially less hygroscopic particles that become more hygroscopic with aging. Regional transport analysis reveals distinct controlling factors: hygroscopicity of nucleation-mode particles is mainly controlled by local sources, while accumulation-mode particles are more influenced by seasonal air mass transport. These results improve understanding of aerosol–cloud interactions and support regional climate modeling and air quality management in urbanizing areas.





## 1. Introduction

Aerosols, defined as mixtures of solid and liquid particles suspended in the atmosphere, possess hygroscopicity as a fundamental physicochemical property that governs their interactions with water vapor under varying environmental relative humidity (RH). This critical property, characterized by the particles' ability to absorb water, leads to a wide range of atmospheric impacts through complex mechanisms (Chen et al., 2019; Zhang et al., 2023).

Aerosol hygroscopicity plays a central role in determining the phase state, chemical reactions, optical properties, and cloud

nucleation activity of aerosol particles, thereby significantly influencing atmospheric chemistry, radiation, and climate effects (Chen et al., 2022; Peng et al., 2020; Ray et al., 2023; Swietlicki et al., 1999; Wang et al., 2025). Hygroscopic growth notably alters aerosol optical properties by modifying particle size distribution and refractive index, enhancing light scattering coefficient (Liu et al., 2022; Song et al., 2023). Furthermore, hygroscopicity promotes cloud formation by activating particles as cloud condensation nuclei, which leads to substantial changes in radiative forcing patterns (Rosenfeld et al., 2014;

Svenningsson et al., 2006). However, aerosol hygroscopicity is influenced by various factors, including environmental conditions and the physicochemical properties of aerosols, resulting in varying hygroscopicity across different environments (Gysel et al., 2007; Jiang et al., 2025).

Currently, multiple instruments and techniques are available for measuring aerosol hygroscopicity. Among the more established instruments are the hygroscopicity-tandem differential mobility analyser (H-TDMA), the cloud condensation

nuclei counter (CCN$_C$), and the dual-nephelometer system (Chen et al., 2023; Jin et al., 2022; Song et al., 2023; Wang et al., 2017; Wang et al., 2023; Zhang et al., 2017). While these instruments can measure aerosol hygroscopicity, their measurement principles differ, and as a result, the resulting hygroscopicity data may show discrepancies (Liu et al., 2021; Liu et al., 2022; Ray et al., 2023; Zhang et al., 2017). Compared to other hygroscopicity measurement instruments, the H-TDMA, which is based on measuring particle number concentration, offers distinct advantages in studying the properties of ultrafine mode

particles (Wang et al., 2019). In addition to measuring the size-resolved hygroscopic growth factor (GF) of aerosols under varying relative humidity conditions, the H-TDMA can also provide insights into the mixing state of particles and reflect aerosol chemical compositions (Chen et al., 2022; Wang et al., 2018; Wang et al., 2019). According to Köhler theory, GF depends on chemical composition (Raoult effect) and particle size (Kelvin effect) (Petters and Kreidenweis, 2007). Compared with GF, hygroscopicity parameter ($\kappa$) introduced by Petters and Kreidenweis (2007) eliminates the influence of the Kelvin

effect, enabling direct comparison of hygroscopicity governed by the Raoult effect across different particle sizes (Kammermann et al., 2010; Petters and Kreidenweis, 2007).

A more profound understanding of aerosol hygroscopicity is crucial for improving the predictive capability of global climate models, particularly in simulating aerosol size distributions and their scattering properties under varying humidity conditions. The size dependence of aerosol hygroscopicity exhibits significant variations across different atmospheric environments (Peng

et al., 2020; Zhang et al., 2023). To better characterize the spatiotemporal evolution of aerosol hygroscopicity under diverse pollution conditions, extensive and comprehensive studies on its spatiotemporal variability are required. Furthermore,



understanding the effects of atmospheric processes, including new particle formation (NPF) and haze events, on aerosol hygroscopicity across heterogeneous environmental regimes remains imperative (Chen et al., 2022).

NPF, defined as the evolution process where newly formed sub–3 nm particles grow to larger sizes, represents a significant
atmospheric aerosol source capable of influencing aerosol hygroscopicity and has been observed globally (Hirshorn et al., 2022; Shen et al., 2023; Yli-Juuti et al., 2011). Generally, freshly nucleated particles exhibit lower hygroscopicity, and the hygroscopicity increases with particle aging (Asmi et al., 2010). NPF events dominated by different chemical components exert distinct impacts on aerosol hygroscopicity. Newly formed sulfate-dominated particles exhibit higher hygroscopicity, whereas organics-dominated counterparts display lower hygroscopicity (Hong et al., 2024; Ma et al., 2016). However, Liu et
al. (2021) observed that on NPF events, photochemical oxidation triggered by the nucleation of volatile organic compounds generates more water-soluble organic acids, resulting in higher hygroscopicity of organic aerosols on NPF days compared to non-NPF days.

In recent years, numerous studies on aerosol hygroscopicity based on H-TDMA observation data have been conducted globally. It has been found that the probability density function of GF or $\kappa$ (GF/$\kappa$-PDF) exhibits a bimodal distribution at urban sites
(Shi et al., 2022; Spitieri et al., 2023; Tan et al., 2013; Wang et al., 2018), while at some non-urban sites, it shows a unimodal or quasi-unimodal distribution (Chen et al., 2022; Wang et al., 2018). Wang et al. (2019) discovered that nucleation-mode particles during clean periods mainly originate from nucleation events followed by growth, whereas during severe pollution periods, they predominantly come from primary emissions in urban environments. Conversely, accumulation-mode particles are primarily from primary emissions during clean periods and secondary processes during pollution periods, resulting in
notable differences of aerosol hygroscopicity for different mode particles under varying pollution levels. Additionally, aerosol hygroscopicity exhibits substantial differences among air masses of varying origins. Over the Antarctic continent, dry continental air masses are reported to exhibit stronger hygroscopicity than moist marine air masses (Asmi et al., 2010). At an urban site in Beijing, seasonal hygroscopicity variations are strongly correlated with air mass source regions (Zhang et al., 2023).

Aerosol concentration and composition undergo significant variations on both temporal and spatial scales. Long-term measurements of aerosol hygroscopicity are crucial for understanding its seasonal and annual variations, as well as its impacts on visibility, atmospheric chemistry, and climate change (Peng et al., 2020). In some regions of the world, studies utilizing H-TDMA for long-term measurements of aerosol hygroscopicity have already been conducted (Alonso-Blanco et al., 2019; Fors et al., 2011; Kammermann et al., 2010; Mamali et al., 2018; Ray et al., 2023). These observations reveal that aerosol
hygroscopicity is highly depending on their sources and physicochemical aging processes. In China, numerous observational experiments measuring aerosol hygroscopicity using H-TDMA have been carried out in regions such as the North China Plain (NCP), the Yangtze River Delta (YRD), and the Pearl River Delta (PRD) (e.g., Jiang et al., 2016; Jiang et al., 2025; Wang et al., 2017). However, most measurements limit to short-term field campaigns (typically 1–2 months duration), which makes it impossible to determine the seasonal variations and their factors in aerosol hygroscopicity at specific locations (Fan et al.,
95  2020).



In this study, the H-TDMA system is utilized in the northern suburbs of Nanjing to obtain size-resolved hygroscopicity observation data for submicron aerosols, covering the entire year from January to December 2021. The H-TDMA observations enable the determination of size-resolved and seasonal variations in aerosol hygroscopicity in the Nanjing region and further facilitate the analysis of influencing factors contributing to these hygroscopicity differences. This paper is structured as follows.

Section 2 describes the instrumentation and the methods to data analysis. Aerosol hygroscopicity during different seasons are discussed in Sect. 3. Conclusions and summary are given in Sect. 4.

## 2. Experiment and data analysis

### 2.1. Measurement site and campaign

A comprehensive field observation experiment was conducted in the northern suburbs of Nanjing in 2021, aiming to delve

deeply into the interactions among the atmosphere, aerosol, boundary layer, and cloud interactions. The observation site is located on the campus of Nanjing University of Information Science and Technology (NUIST, 32°13′ N, 118°46′ E) in the northern suburban area of Nanjing, which is situated in the central YRD. Further details on the field campaign and measurement site are available in Song et al. (2023).

This work provides a comprehensive examination of seasonal variations in aerosol hygroscopicity and their relationship with

chemical composition, based on year-round observational data collected from January to December 2021. The dataset is partitioned according to conventional meteorological seasons: winter (Jan–Feb and Dec 2021), spring (Mar–May 2021), summer (Jun–Aug 2021), and autumn (Sep–Nov 2021).

### 2.2. Measurements and data analysis

#### 2.2.1. Measurement of aerosol hygroscopicity

The H-TDMA used in this study primarily consists of two differential mobility analysers (DMA, Model 3081L, TSI Inc.) and a condensation particle counter (CPC, Model 3772, TSI Inc.). The dried and neutralized aerosol sample is first passed through the first DMA, which selects monodisperse particles of specific diameters (40, 80, 110, 150, and 200 nm). Subsequently, the sample with monodisperse particles is humidified to RH=90% via a nafion humidifier. Finally, the sample is directed through the second DMA and the CPC to measure the particle number size distribution of the humidified particles.

The hygroscopic growth factor (GF) is defined as:

$$\mathrm{GF} = \frac{D_\mathrm{p}(\mathrm{RH})}{D_\mathrm{p}(\mathrm{dry})}, \tag{1}$$

where $D_\mathrm{p}(\mathrm{dry})$ denotes the dry diameter of monodisperse particles selected by the first DMA before humidification, and $D_\mathrm{p}(\mathrm{RH})$ represents the particle diameter selected by the second DMA after humidification at RH = 90%.



The H-TDMA data can be used to compute the measured distribution function of GF for any selected $D_p$(dry) particles, which is then used to retrieve the probability distribution function of GF (GF-PDF) according to the multi-mode TDMAfit algorithm. The hygroscopicity parameter ($\kappa$) is calculated as follows (Petters and Kreidenweis, 2007):

$$\kappa = (GF^3 - 1) \cdot [\frac{1}{RH} \exp\left(\frac{4\,\sigma_{s/a}\,M_w}{R\,T\,\rho_w\,D_p\,GF}\right) - 1]\,, \tag{2}$$

where $\sigma_{s/a}$ is the surface tension of the droplet–air interface at the composition of the droplet, $M_w$ is the molar mass of water, $R$ is the universal gas constant, $T$ is the temperature, and $\rho_w$ is the density of water.

The probability distribution function of $\kappa$ ($\kappa$-PDF, $c\,(\kappa, D_p)$) derived from GF-PDF is normalized by $\int c\,(\kappa, D_p)\,d\kappa = 1$. Based on the $\kappa$ values, aerosol particles are categorized into three hygroscopic groups: nearly hydrophobic (NH, $\kappa < 0.1$), less hygroscopic (LH, $0.1 \leq \kappa < 0.2$), and more hygroscopic (MH, $\kappa \geq 0.2$). The MH groups mainly consist of inorganic species such as sulfates, nitrates, and ammonium salts. In contrast, the NH and LH groups are primarily composed of black carbon, insoluble organics, and partially soluble organics (Liu et al., 2011; Müller et al., 2017).

The mean $\kappa$ ($\kappa_{mean}$) is then defined as the number-weighted mean of $\kappa$-PDF over the $\kappa$ range $[a, b]$:

$$\kappa_{mean} = \int_a^b \kappa\, c\,(\kappa, D_p)\,d\kappa\,, \tag{3}$$

where $a$ and $b$ represent the lower and upper integration limits of $\kappa$. For the ensemble $\kappa_{mean}$, the integral spans the entire domain, i.e., $a = 0$ and $b \to \infty$. When computing $\kappa_{mean}$ of NH, LH, and MH hygroscopic groups ($\kappa_{NH}$, $\kappa_{LH}$, and $\kappa_{MH}$), $a$ and $b$ are set according to the predefined $\kappa$ range of each group.

Accordingly, the number fraction (NF) for each hygroscopic group with over the range $[a, b]$ is defined as:

$$NF = \int_a^b c\,(\kappa, D_p)\,d\kappa\,. \tag{4}$$

The standard deviation of $\kappa$-PDF ($\sigma_{\kappa\text{-PDF}}$) is calculated as:

$$\sigma_{\kappa\text{-PDF}} = \left(\int_0^\infty (\kappa - \kappa_{mean})^2\, c\,(\kappa, D_p)\,d\kappa\right)^{\frac{1}{2}}\,. \tag{5}$$

Previous research has widely adopted $\sigma_{\kappa\text{-PDF}}$ as a metric for aerosol mixing state characterization (Jiang et al., 2016; Wang et al., 2017). However, this absolute dispersion parameter fails to account for scenarios where significant $\kappa_{mean}$ variations exist between different sizes. To address this limitation, this study introduces the coefficient of variation ($CV_{\kappa\text{-PDF}}$):

$$CV_{\kappa\text{-PDF}} = \frac{\sigma_{\kappa\text{-PDF}}}{\kappa_{mean}}\,. \tag{6}$$

This normalized parameter effectively captures the relative dispersion of size-resolved $\kappa$-PDF by incorporating $\kappa_{mean}$ differences among particle populations. Consequently, $CV_{\kappa\text{-PDF}}$ serves as the primary mixing state indicator in this study, with lower values corresponding to stronger internal mixing.





### 2.2.2. Measurements of other aerosol properties

The measurement of aerosol chemical compositions is conducted using an aerosol chemical speciation monitor (ACSM, Aerodyne Research Inc.) coupled with an aethalometer (AE-33, Magee Scientific Inc.). Both instruments are configured with $PM_{2.5}$ sampling inlets: the ACSM system incorporates a $PM_{2.5}$ aerodynamic lens, while the AE-33 utilizes an aerosol cutter
with 2.5 μm cutoff diameter. This configuration ensures measurement of particulate matter mass concentrations specifically for the aerodynamic diameter fraction below 2.5 μm ($PM_{2.5}$).

The ACSM operates at a temporal resolution of 15 minutes, while the AE-33 provides measurements at a higher resolution of 1 minutes. The ACSM, equipped with a capture vaporizer and a quadrupole mass spectrometer, is primarily used to measure the mass concentrations of organic aerosols (OA), sulfate ($SO_4^{2-}$), nitrate ($NO_3^-$), ammonium ($NH_4^+$), and chloride (Chl). To
further analyse OA composition, the positive matrix factorization analysis is applied to differentiate between primary organic aerosols (POA) and secondary organic aerosols (SOA). Concurrently, the AE-33 is employed to measure the mass concentration of black carbon (BC). The total $PM_{2.5}$ mass concentration is derived by summing the mass concentrations of all components measured by the ACSM and AE-33. The absence of dust-related measurements may lead to underestimation of $PM_{2.5}$ mass concentrations during dust episodes in this study.

The particle number size distribution (PNSD) is measured by two scanning mobility particle sizers (SMPS) covering different size ranges: Nano-SMPS (2–60 nm) and SMPS (15–700 nm), both operating with a temporal resolution of 5 minutes. The Nano-SMPS is specifically used for auxiliary identification of NPF events. Particles are classified into two modes based on their diameter: nucleation mode ($D_p \leq 100$ nm) and accumulation mode (100 nm $< D_p \leq 700$ nm). Then the total number concentrations in nucleation and accumulation modes ($N_{nuc}$ and $N_{acc}$) are then calculated.

**2.3. Backward trajectory calculation and clustering analysis**

Backward trajectories of air masses arriving at the sampling site are calculated using the NOAA HYSPLIT-4 (Hybrid Single-Particle Lagrangian Integrated Trajectory) model (Draxler and Hess, 1998; Wu et al., 2016). The 72-hour backward trajectories are initiated at 3-hour intervals from 00:00 to 21:00 LT (Local Time, UTC+08) and terminated at 100 m above ground level. The number of clusters is determined based on the variation in total spatial variance (refer to the HYSPLIT4 User Guide), with
the k-means clustering algorithm applied to classify trajectories for each season. To balance optimal trajectory separation (favoring a larger number of clusters) with visualization simplicity (preferring fewer clusters), air mass backward trajectories are partitioned into 3 clusters per season.





# 3. Results and discussion

## 3.1. Overview

### 3.1.1. Seasonal variations of size-resolved aerosol hygroscopicity

As shown in Figure 1a, a significant increase in $\kappa_{mean}$ with particle size is observed in all seasons, consistent with a previous study in Nanjing (Jiang et al., 2025). The size-resolved annual $\kappa_{mean}$ values are $0.12 \pm 0.04$, $0.16 \pm 0.05$, $0.18 \pm 0.05$, $0.20 \pm 0.05$, and $0.22 \pm 0.06$ for particles with diameters of 40, 80, 110, 150, and 200 nm, respectively (Table S1). Similar hygroscopicity-particle size dependence is observed in Madrid, while more complex relationships are found in Athens, India and Switzerland (Alonso-Blanco et al., 2019; Spitieri et al., 2023; Ray et al., 2023; Kammermann et al., 2010). The enhancement of aerosol hygroscopicity with particle size may be attributed to chemical aging processes of particles and an increased proportion of inorganic components (Alonso-Blanco et al., 2019; Wu et al., 2016). Compared to the measurements in the NCP (Chen et al., 2022), $\kappa$ at this site is significantly lower, likely due to the dominance of organic aerosols (annual average mass fraction: 42.92% in PM$_{2.5}$) (Figure 1b). This pattern closely resembles observations from Shanghai, which is also located within the YRD region (Chen et al., 2022).

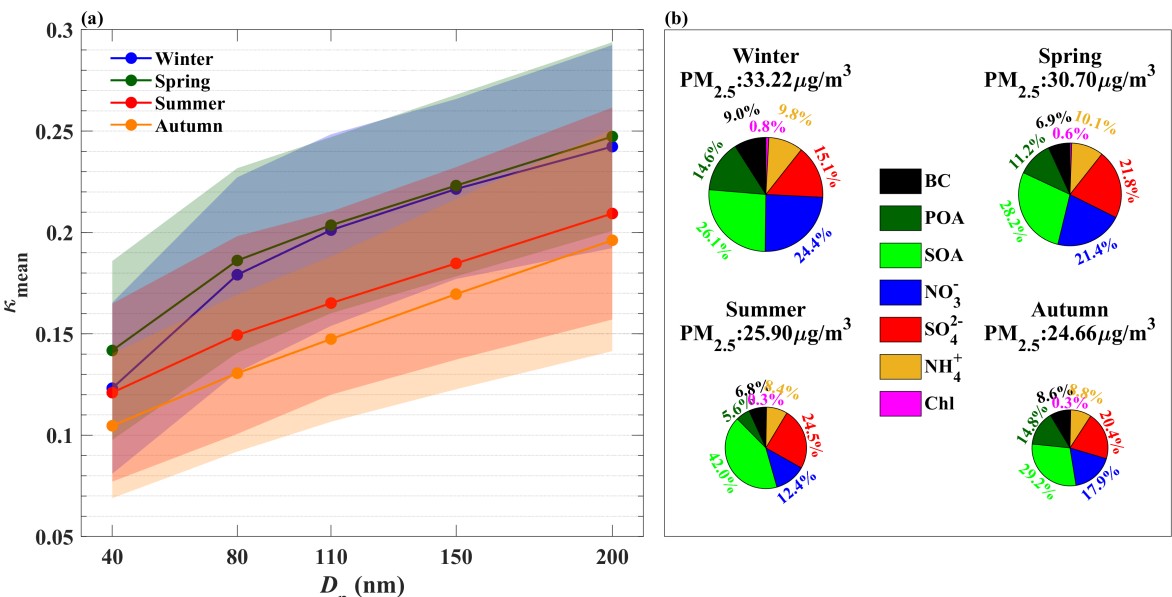

**Figure 1. (a) Size variations of the mean hygroscopicity parameter ($\kappa_{mean}$) during different seasons, with error bands indicating the standard deviations of $\kappa_{mean}$. (b) Seasonal distributions of mass concentrations and compositional fractions for PM$_{2.5}$ chemical species.**





A pronounced seasonal contrast in $\kappa_{mean}$ is observed for 40 nm particles, with spring exhibiting the highest value ($\kappa_{mean} = 0.14$) compared to other seasons ($\kappa_{mean} = 0.10–0.12$). For larger particles (80–200 nm), obviously higher $\kappa_{mean}$ values are recorded in winter and spring than in summer and autumn. This pattern may be attributed to the increased mass fractions of more hygroscopic sulfate–nitrate–ammonium (SNA) during the more severe PM$_{2.5}$ pollution in winter and spring. As shown in

Figure 1b, the mass concentrations of PM$_{2.5}$ during winter and spring exceed 30 μg m$^{-3}$, with corresponding higher MF$_{SNA}$ (the mass concentration of SNA, MF$_{SNA}$) values of 49.37% and 53.25%, respectively. In contrast, summer and autumn measurements show lower PM$_{2.5}$ levels, averaging approximately 25 μg m$^{-3}$, with corresponding lower MF$_{SNA}$ values of 45.33% and 47.13%, respectively. Similar seasonal variation patterns in aerosol chemical composition are reported at another site in Nanjing during 2016–2017 (Xie et al., 2022).

Figure 1a also suggests that nucleation-mode particles (40 and 80 nm) display greater size-dependent $\kappa_{mean}$ variability than accumulation-mode particles (110, 150, and 200 nm), particularly during winter and spring. This is likely due to the higher organic content in nucleation-mode particles, especially in winter and spring. Consequently, the $\kappa_{mean}$ increase rate with particle size is substantially steeper across the nucleation mode than the accumulation mode size range.

In summary, aerosols in the northern suburban area of Nanjing exhibit relatively low hygroscopicity, primarily due to their

elevated organic content. A consistent increase in hygroscopicity with particle size is observed across all seasons, with this trend being particularly pronounced for nucleation-mode particles (40–80 nm). Furthermore, aerosols in winter and spring demonstrate enhanced hygroscopicity in the 40–200 nm size range compared to summer and autumn, which can be attributed to higher concentrations of SNA.

### 3.1.2. Seasonal variations of hygroscopic groups in the different size particles

Figure 2a illustrates the seasonal variations in the NF and $\kappa$ of different hygroscopic groups (NH, LH, MH) across various particle sizes. With the exception of 40 nm particles, the size variations in $\kappa_{NH}$ and $\kappa_{LH}$ are less pronounced than those of $\kappa_{MH}$. Meanwhile, NF$_{MH}$ shows a notable increase with particle size. Considering the increase of $\kappa_{mean}$ with particle size (Figure 1a), this phenomenon suggests that the size variation in $\kappa_{mean}$ is mostly driven by MH group particles. Unlike particles in other size ranges, the 40 nm particles have relatively small seasonal differences in $\kappa_{NH}$, $\kappa_{LH}$, and $\kappa_{MH}$. The $\kappa_{mean}$ of 40 nm particles in

spring is much higher than in other seasons due to the lower NF$_{NH}$ and higher NF$_{MH}$ of 40 nm particles.

For nucleation-mode particles, $\kappa_{NH}$ is found to increase consistently from winter and spring to summer and autumn, whereas for accumulation-mode particles, the lowest $\kappa_{NH}$ values are recorded during summer. However, the variation in $\kappa_{NH}$ remains minor, generally within the range of 0.03–0.05. In contrast, $\kappa_{LH}$ for nucleation-mode particles is slightly higher in spring compared to other seasons, while for accumulation-mode particles, it reaches slightly higher values in summer. Despite these

variations, $\kappa_{LH}$ remains stable, typically ranging between 0.14–0.17, which is in the $\kappa$ range of SOA (Petters and Kreidenweis, 2007). Across all particle sizes, both $\kappa_{MH}$ and NF$_{MH}$ are bigger in winter and spring than in summer and autumn for the same particle sizes. Considering the higher $\kappa_{mean}$ in winter and spring compared to summer and autumn (Figure 1a), this suggests that the seasonal in $\kappa_{mean}$ is likely driven by MH group particles.



In general, aerosol particles in winter exhibit higher $\kappa_{MH}$ compared to spring likely due to their greater abundance of $NO_3^-$, which possesses stronger hygroscopicity. This effect is particularly evident in accumulation-mode particles, which displays a more pronounced seasonal contrast. However, a significantly higher $NF_{MH}$ is observed in spring compared to winter due to the greater abundance of SNA, leading to an increased $\kappa_{mean}$ in spring (Figure 1). Although the differences in aerosol $\kappa_{MH}$ between summer and autumn are relatively small, the higher $NF_{MH}$ in summer results in greater $\kappa_{mean}$ across all particle size ranges.

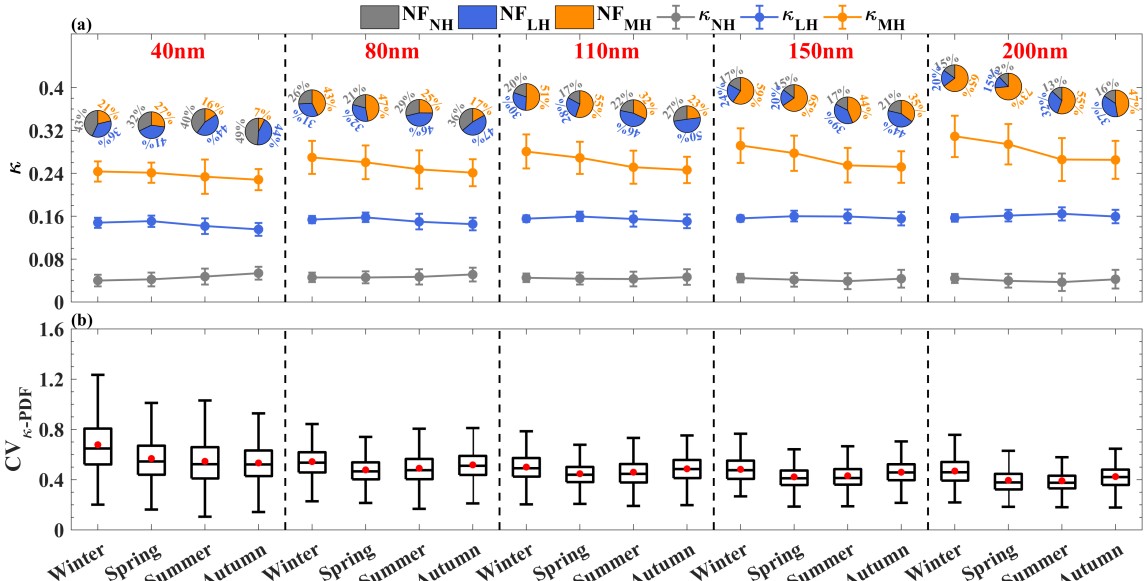

**Figure 2. (a) Seasonal variations of $NF_{MH}$, $NF_{LH}$, and $NF_{NH}$ (represented by orange, blue, and gray segments in the pie charts, respectively), alongside the corresponding $\kappa_{MH}$, $\kappa_{LH}$, and $\kappa_{NH}$ (represented by orange, blue, and gray lines) for different size particles (40–200 nm). Error bars indicate the standard deviation of each parameter. (b) Seasonal variations of $CV_{\kappa\text{-PDF}}$ for different size particles. Boxplots display the mean (red dots), interquartile range (25th–75th percentiles), and 5th–95th percentile ranges.**

Figure 2b demonstrates a consistent inverse relationship between $CV_{\kappa\text{-PDF}}$ and particle size across all seasons, reflecting enhanced internal mixing during aerosol aging and growth processes. Notably, nucleation-mode particles exhibit significantly higher $CV_{\kappa\text{-PDF}}$ values in winter compared to other seasons, suggesting a more pronounced external mixing state. This seasonal pattern likely results from reduced photochemical activity during winter months, which inhibits atmospheric aging processes and promotes the persistence of externally mixed aerosols.

To further elucidate the impact of different hygroscopic groups on $\kappa_{mean}$, the correlation coefficients ($R^2$) between the $\kappa_{mean}$ and six parameters ($NF_{MH}$, $NF_{LH}$, $NF_{NH}$, $\kappa_{MH}$, $\kappa_{LH}$, and $\kappa_{NH}$) are calculated (Figure 3). The results indicate that $\kappa_{mean}$ is predominantly influenced by $NF_{MH}$ in all seasons. For nucleation-mode particles, $\kappa_{mean}$ is also affected largely by $NF_{NH}$, particularly for 40 nm particles, where the $R^2$ between $NF_{NH}$ and $\kappa_{mean}$ (0.76–0.89) is even slightly higher than that between




NF$_{MH}$ and $\kappa_{mean}$ (0.66–0.88). This phenomenon can be attributed to the fact that nucleation-mode particles, primarily originating from direct emissions or NPF events with limited aging, are typically composed of more hydrophobic matters (e.g., OA and BC) (Gysel et al., 2007; Li et al., 2023). These particles are characterized by higher NF$_{NH}$ and lower NF$_{MH}$, leading to NF$_{NH}$ being a dominant factor in determining $\kappa_{mean}$ for nucleation-mode particles.



**Figure 3.** The correlation coefficients (R2) between κmean and six parameters (NF$_{MH}$, NF$_{LH}$, NF$_{NH}$, $\kappa_{MH}$, $\kappa_{LH}$, $\kappa_{NH}$) for different size particles (40–200 nm) during different seasons.

Furthermore, these findings suggest that variations in the NF of different hygroscopic groups have a greater impact on $\kappa_{mean}$ than variations in the $\kappa$ values of these groups. Beyond the influence by the NF of different hygroscopic groups, $\kappa_{mean}$ for accumulation-mode particles is also significantly affected by $\kappa_{MH}$, while for nucleation-mode particles, due to their relatively





weaker hygroscopicity, $\kappa_{\text{mean}}$ is jointly influenced by $\kappa_{\text{MH}}$ and $\kappa_{\text{LH}}$. Compared to winter and spring, the influence of $\kappa_{\text{MH}}$ on $\kappa_{\text{mean}}$ is smaller in summer and autumn, whereas the influence of $\kappa_{\text{LH}}$ on $\kappa_{\text{mean}}$ is greater. This pattern is attributed to the $\kappa_{\text{mean}}$ across different particle sizes is lower in summer and autumn than in winter and spring.

### 3.1.3. Seasonal variation of diurnal cycles in aerosol hygroscopicity

As illustrated in Figure 4a1–a4, rush-hour-induced enhancements of particle number concentration demonstrate distinct seasonal variability, with particularly pronounced effects on nucleation-mode particles during morning (~06:00 LT) and evening (~20:00 LT) periods. The phenomenon attains its maximum intensity during winter and spring seasons, periods that frequently coincide with atmospheric stagnation events. Quantitative analysis reveals substantial PM$_{2.5}$ mass accumulation during rush hours across all non-summer seasons, accompanied by synchronous increases in both BC mass concentrations and fractions ($M_{\text{BC}}$ and MF$_{\text{BC}}$) (Figure 4). These traffic-induced compositional changes in rush hours drive systematic reductions in $\kappa_{\text{mean}}$ values across all particle size ranges due to increased NF$_{\text{NH}}$ (Figure 5). Notably, the suppression effect on particle hygroscopicity exhibits its greatest sensitivity to traffic emissions during summer months.

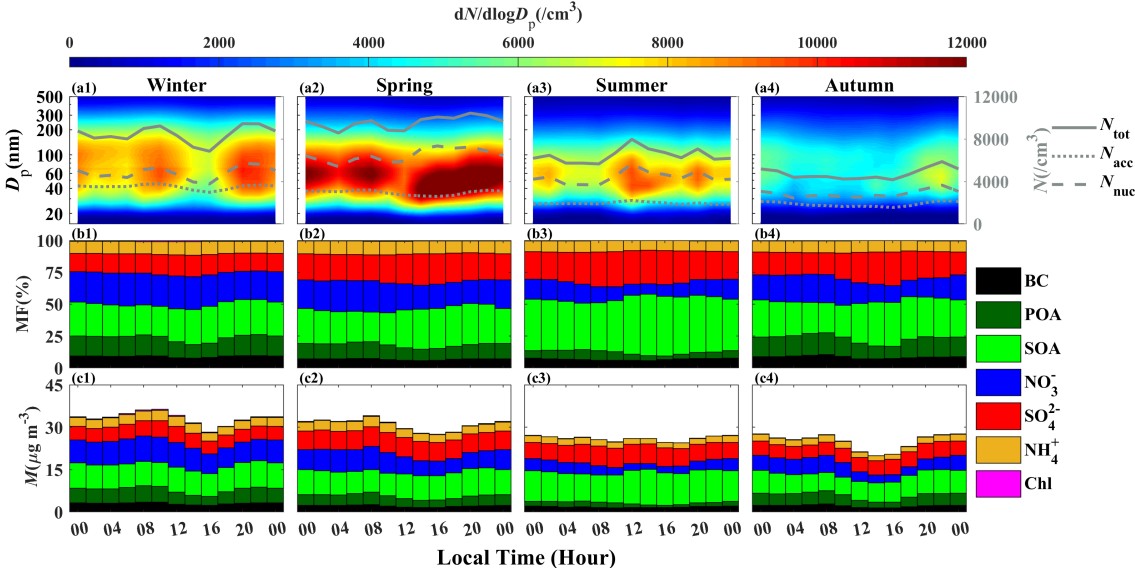

**Figure 4. Diurnal variations of (a1–a4) particle number size distributions (PNSD), total particle number concentration ($N_{\text{tot}}$), accumulation-mode particle number concentration ($N_{\text{acc}}$), and nucleation-mode particle number concentration ($N_{\text{nuc}}$), (b1–b4) mass fractions (MF) of aerosol chemical species, and (c1–c4) mass concentrations ($M$) of aerosol chemical species during different seasons.**





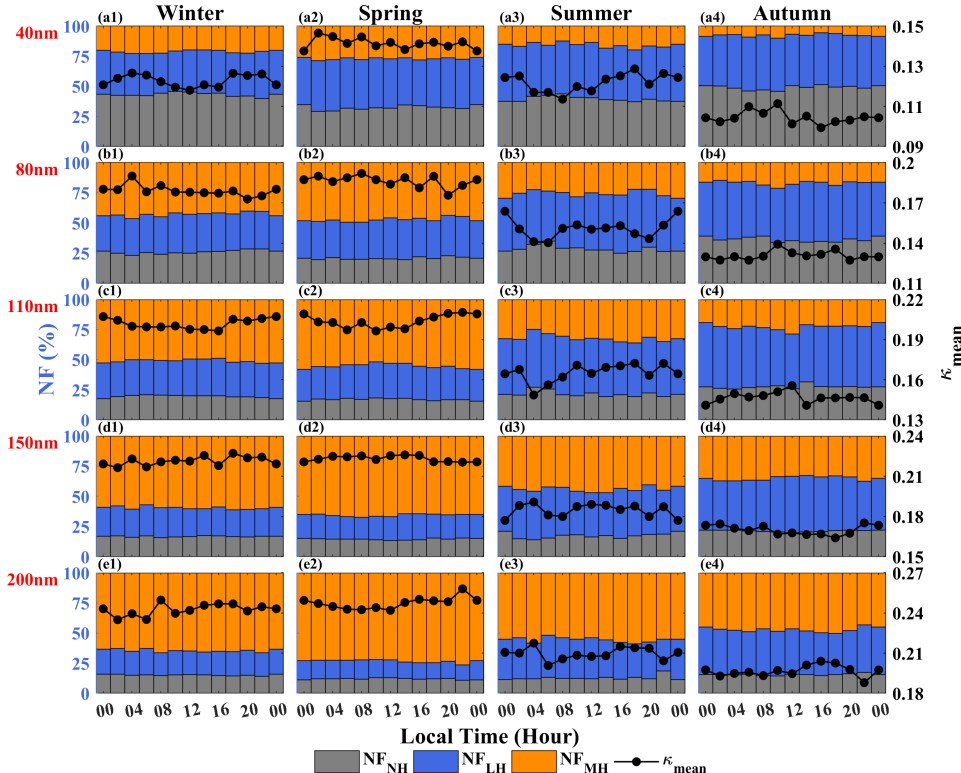


**Figure 5. Diurnal variations of the NF$_{MH}$ (orange bars), NF$_{LH}$ (blue bars), and NF$_{NH}$ (gray bars), and $\kappa_{mean}$ (black dots) for different size particles (40–200 nm) during different seasons.**

Figure 4b1–b4 reveals pronounced daytime enhancements in the mass fractions of SO$_4^{2-}$ and SOA mass fractions across all

seasons, with peak values coinciding with periods of maximum photochemical activity—a trend most pronounced in summer

due to enhanced solar radiation (Peng et al., 2017). In contrast, the mass fraction of NO$_3^-$ displays an inverse diurnal cycle,

reaching maximal mass fractions during nocturnal periods. This behavior is mechanistically explained through temperature-

mediated phase partitioning theory, where lower nocturnal temperatures coupled with higher relative humidity facilitate

efficient gas-to-particle conversion through aqueous-phase oxidation processes (Sun et al., 2013).

These compositional shifts drive distinct hygroscopicity dynamics in accumulation-mode particles. As illustrated in Figure 5,

the $\kappa_{mean}$ for 200 nm particles displays a bimodal diurnal pattern due to increased NF$_{MH}$: a primary afternoon peak (14:00–

18:00 LT), driven by photochemical production of hydrophilic SO$_4^{2-}$ and SOA, and a secondary evening enhancement (post-

20:00 LT), associated with NO$_3^-$ accumulation under favorable nighttime chemical conditions.





## 3.2. Impact of NPF events on the size-resolved aerosol hygroscopicity

Figure 4a2 indicates that spring exhibits distinctive particle dynamics, characterized by frequent NPF events. The PNSD pattern displays a unique banana-shaped diurnal cycle, with the frequency of NPF occurrences reaching 20.65% during spring—approximately double the annual average and significantly higher than in other seasons (Table S2). Previous studies have shown that strong winds during prevalent spring dust episodes in China significantly scavenge fine particles, creating a cleaner environment conducive to NPF occurrence (Shen et al., 2023). In contrast to spring non-NPF days, the spring NPF

days shows more pronounced diurnal fluctuations in the number concentration of nucleation-mode particles and the mass concentration of PM$_{2.5}$ chemical compositions (Fig. S1). As shown in Fig. S1c1, around 10:00 LT in the spring NPF days, the mass concentration of PM$_{2.5}$ sharply decreases. Under these low pollution conditions, the lower condensation sink (CS) facilitates the onset of NPF events (Hong et al., 2023). This leads to a rapid increase in $N_{nuc}$ (Fig. S1a1).

In comparison to spring non-NPF days, MF$_{OA}$ significantly increases in spring NPF days, with MF$_{POA}$ gradually decreases,

and MF$_{SOA}$ gradually increases over time. This trend starts at 10:00 LT and continues until 18:00 LT to the evening rush hours (Fig. S1b1). This suggests that spring NPF events are predominantly driven by the formation of SOA. Additionally, a markedly lower $\kappa_{mean}$ is observed for 40 nm particles (Fig. S2a1), suggesting that the newly formed nucleation-mode particles during NPF events exhibits lower hygroscopicity. This is related to a higher NF$_{NH}$ and a lower NF$_{MH}$ (Fig. S2b1 and Fig. S2d1). As the nucleation-mode particles grows, the $\kappa_{mean}$ of 40 nm particles slightly increases, although it remains lower than on non-

NPF days (Fig. S2).

Between 14:00–16:00 LT, the $N_{nuc}$ reaches its peak, while the $N_{acc}$ and the PM$_{2.5}$ mass concentration reach their diurnal lows (Fig. S1). A distinct diurnal peak in $\kappa_{mean}$ occurs around 16:00 LT for all size particles, driven by concurrent increases in both the NF$_{MH}$ and $\kappa_{MH}$ (Figs. S2 and S3). This enhancement is particularly evident for accumulation-mode particles, which have a higher NF$_{MH}$. These findings suggest that newly formed nucleation-mode particles, which exhibit lower hygroscopicity,

gradually age and grow into more hygroscopic accumulation-mode particles over time during spring NPF days.

As shown in Figure 6a, the $\kappa_{NH}$, $\kappa_{LH}$ and $\kappa_{MH}$ exhibit minimal difference between NPF days and non-NPF days during spring. However, the $\kappa_{mean}$ on NPF days is significantly lower by 6.76% for 40 nm particles and higher by 5.07% for 200 nm particles compared to non-NPF days (Table S3). This divergence is attributed to the changes in the NF$_{NH}$ and NF$_{MH}$: NPF days have larger NF$_{NH}$ and smaller NF$_{MH}$ for 40 nm particles, while the opposite is observed for 200 nm particles (Figure 6b). Additionally,

the diurnal variation of CV$_{\kappa\text{-PDF}}$ for 40 nm particles in NPF days has obvious fluctuation, while the CV$_{\kappa\text{-PDF}}$ for other particle sizes in NPF days and all particle sizes in non-NPF days has no obvious diurnal variation. This indicates that the NPF event only significantly affects the internal mixing state of small particles (40 nm particles). Furthermore, the CV$_{\kappa\text{-PDF}}$ for nucleation-mode particles on NPF days is slightly larger than on non-NPF days, while the opposite is observed for accumulation-mode particles (Table S3). These findings suggest that newly formed particles during NPF events exhibit reduced internal mixing

compared to non-NPF conditions, while aged larger particles demonstrate enhanced mixing homogeneity.




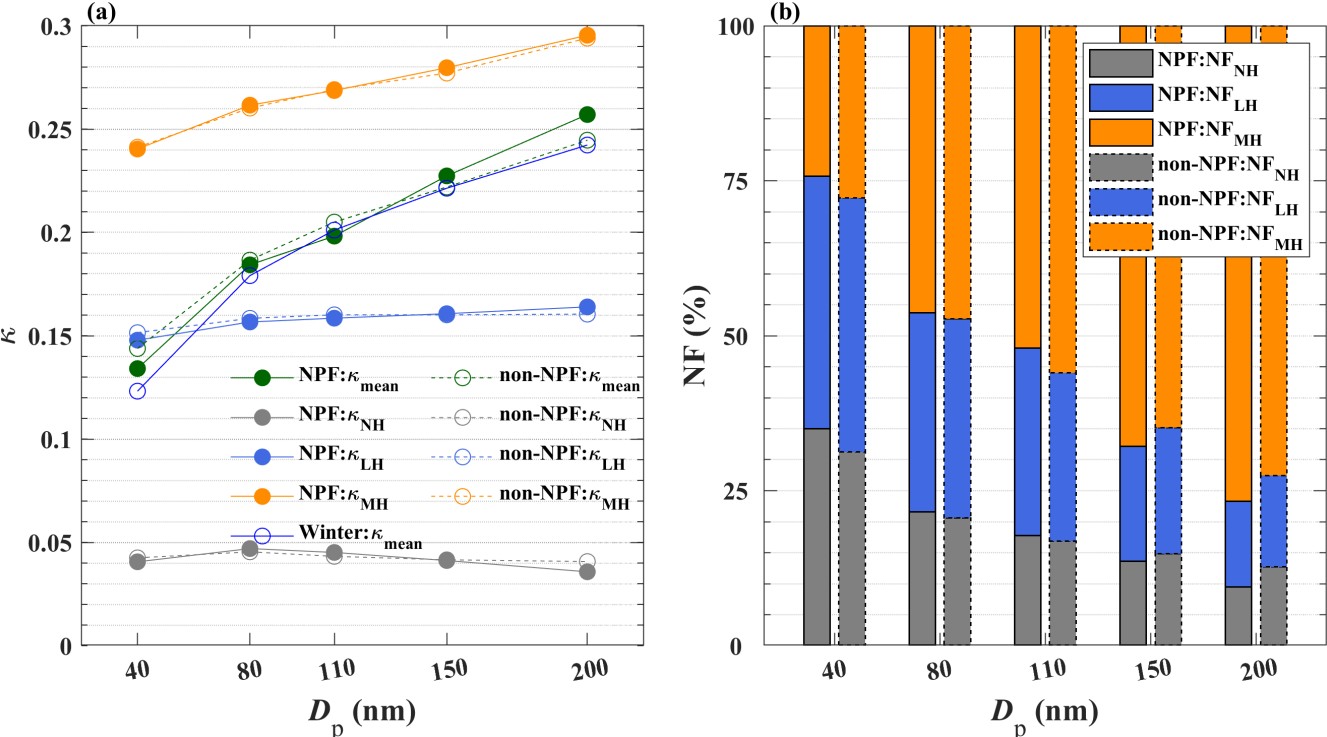

**Figure 6. (a) $\kappa_{mean}$, $\kappa_{MH}$, $\kappa_{LH}$ and $\kappa_{NH}$ in spring NPF days and spring non-NPF days, and the $\kappa_{mean}$ in winter for all size particles (40–200 nm). (b) $NF_{MH}$, $NF_{LH}$, $NF_{NH}$ in spring NPF days and spring non-NPF days for all size particles (40–200 nm).**


Similar with spring, winter exhibits the higher $\kappa_{mean}$ values for particles in the 40-200 nm size range (Figure 1a). To investigate seasonal differences, the $\kappa_{mean}$ values of spring NPF days and non-NPF days are compared with winter conditions (Figure 6a). For nucleation-mode particles, winter consistently shows lower $\kappa_{mean}$ values than both spring NPF and non-NPF days. For accumulation-mode particles (except for 110 nm), winter $\kappa_{mean}$ values are significantly lower than spring NPF days but

comparable to spring non-NPF days, with only marginal differences observed. The 110 nm particles, representing a transitional size between nucleation and accumulation modes, demonstrate size-dependent behavior: during spring NPF events, their hygroscopic properties align more closely with nucleation-mode particles, resulting in lower $\kappa_{mean}$ values compared to winter. In contrast, on spring non-NPF days, 110 nm particles show slightly enhanced hygroscopicity relative to winter conditions. These findings collectively demonstrate that spring NPF events serve to reduce seasonal differences in $\kappa_{mean}$ for nucleation-

mode particles between spring and winter, and enhance the hygroscopicity of accumulation-mode particles specifically in spring.



### 3.3. Impact of regional transport on aerosol hygroscopicity

### 3.3.1. Seasonal impacts of regional transport on aerosol hygroscopicity

Under the influence of the subtropical monsoon, seasonal variations in air mass sources are evident: while summer air masses
derive from southern mountainous areas, other seasons primarily receive air masses from northern plains (Figure 7). Aerosol chemical composition analysis in different clusters (Fig. S4) shows that summer air masses contain substantially higher proportions of SOA in $PM_{2.5}$ compared to other seasons. This compositional difference leads to elevated $NF_{LH}$ values in summer, resulting in significantly lower $\kappa_{mean}$ values for accumulation-mode particles relative to winter and spring (Figure 8). Notably, springtime $PM_{2.5}$ mass concentrations show significant variation among air mass categories, with C2 exhibiting
substantially lower concentrations than C1 and C3 (Fig. S4). As indicated in Table S4, the occurrence frequencies of NPF events during spring for C1, C2, and C3 are 11.16%, 42.44%, and 25.00%, respectively, demonstrating that NPF events in Nanjing preferentially occur under cleaner atmospheric conditions (lower $PM_{2.5}$ mass concentration in C2) in spring.

In contrast, autumn air masses exhibit higher POA content in $PM_{2.5}$ (Fig. S4), yielding comparable $NF_{LH}$ but increased $NF_{NH}$ relative to summer conditions, along with marginally reduced $\kappa_{mean}$ values for equivalent particle sizes (Figure 8). While winter
air masses demonstrate higher $\kappa_{MH}$ (Fig. S5), spring air masses contain greater amounts of SNA in $PM_{2.5}$ (Fig. S4) and the elevated $NF_{MH}$ ultimately produces slightly higher $\kappa_{mean}$ values compared to that in winter (Figure 8). Collectively, these observations reveal that similar particle hygroscopicity under winter–spring air masses, and comparable hygroscopic properties under summer–autumn air masses (Figure 8).





**Figure 7. 72-hour air mass backward trajectories at a height of 100 meters corresponding to the cluster analysis during different seasons. The line colors denote different clusters, i.e., red for Cluster C1, green for Cluster C2, and blue for Cluster C3.**



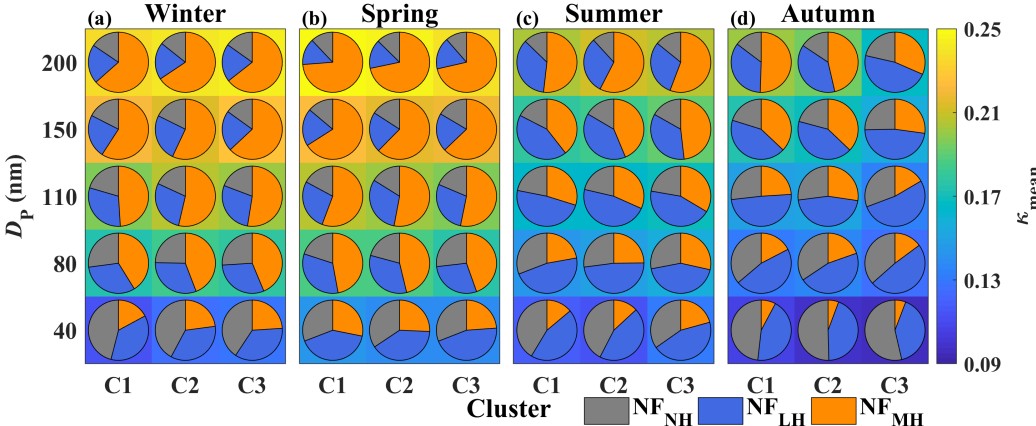

**Figure 8. The $\kappa_{mean}$ (color block) of aerosols and the number fractions of different hygroscopic groups for all size particles (40–200 nm) in the influence of different air masses during different seasons. The color blocks of pie chart denote number fractions of different hygroscopic groups, i.e., orange block of pie chart for $NF_{MH}$, blue block of pie chart for $NF_{LH}$, and gray block of pie chart for $NF_{NH}$.**

### 3.3.2. Influence of regional transport on hygroscopicity of different mode particles

Figure 8 also demonstrates that, across all seasons except autumn, accumulation-mode particles under different air masses exhibit minimal NF variations among hygroscopicity groups within the same season. Although notable differences in $\kappa_{MH}$ are observed at certain particle sizes (Fig. S5), $\kappa_{mean}$ remains relatively consistent. In contrast, accumulation-mode particles during autumn display significant $\kappa_{mean}$ variations across hygroscopic groups, attributable to larger NF differences despite comparable $\kappa_{MH}$ values between air masses. These findings suggest that NF serves as the dominant factor governing $\kappa_{mean}$, a conclusion that aligns with the results presented in Sect.3.1.2.

Integrative examination of Figures Figure 7, Figure 8, and S4 further reveals that, within a given season, accumulation-mode particles originating from eastern/northern air masses consistently exhibit stronger hygroscopicity compared to those from western/southern sources. This enhanced hygroscopic behavior can be attributed to their higher SNA content in $PM_{2.5}$.

For nucleation-mode particles (particularly at 40 nm), $\kappa_{mean}$ is predominantly influenced by $NF_{NH}$ rather than $NF_{MH}$ (Figure 3). Given that $\kappa_{NH}$ remains relatively small and stable, seasonal variations in $\kappa_{mean}$ are negligible despite substantial $NF_{NH}$ differences across air masses (Figure 8). This implies that the hygroscopicity of nucleation-mode particles is less sensitive to air mass origins and is primarily regulated by local sources. However, an exception occurs in spring, where air masses yield significantly higher $\kappa_{mean}$ values for 40 nm particles, a consequence of their elevated $NF_{MH}$ (Figure 8).

Figure 9 shows the $CV_{\kappa\text{-PDF}}$ for size-resolved particles under varying air mass influences across seasons. Comparative analysis reveals that during non-summer seasons, 200 nm particles affected by long-range transported air masses (winter C3, spring C2, summer C2, autumn C3) demonstrate significantly higher $CV_{\kappa\text{-PDF}}$ values relative to other air masses, suggesting more



pronounced external mixing. Interestingly, summer exhibits a distinct pattern where C2-influenced 200 nm particles show even lower $CV_{\kappa\text{-PDF}}$ values compared to summer C1 and C3 cases, indicating more advanced internal mixing. This seasonal anomaly can be attributed to enhanced photochemical aging during summer transport, driven by more intensive solar radiation. Consequently, while C2-associated 200 nm particles display marginally increased hygroscopicity during summer (Figure 8c), other seasons show no statistically significant hygroscopicity enhancement in long-range transported particle clusters.

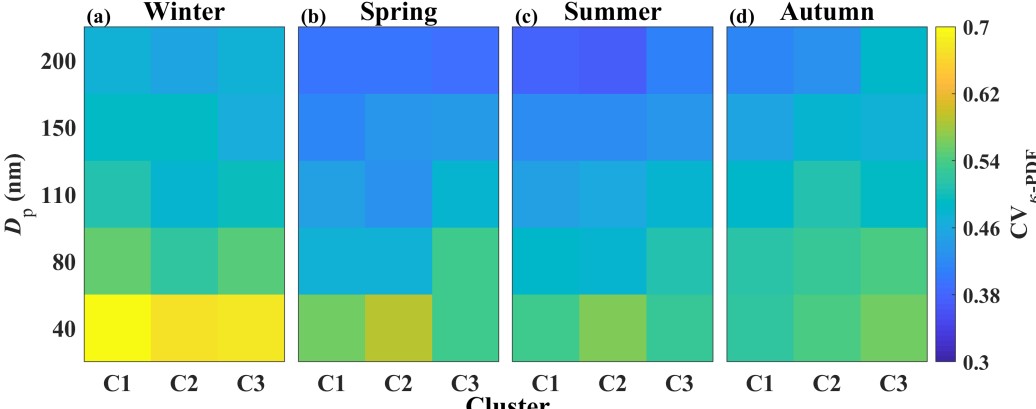

**Figure 9.** The $CV_{\kappa\text{-PDF}}$ for all size particles (40–200 nm) in the influence of different air masses during different seasons.

All these results reveal significantly smaller intra-seasonal hygroscopicity variations compared to inter-seasonal differences. This pattern primarily stems from more pronounced chemical composition disparities between seasonal air masses, which amplify contrasts within strongly hygroscopic particle groups. The study demonstrates distinct spatial controls on particle hygroscopicity: nucleation-mode particles exhibit predominantly local regulation, as evidenced by minimal $\kappa_{\text{mean}}$ variations despite substantial NF fluctuations. In contrast, within a given season, accumulation-mode particles originating from eastern/northern air masses display enhanced hygroscopicity relative to their western/southern counterparts, attributable to elevated SNA concentrations. Notably, long-range air mass transport induces hygroscopicity enhancement exclusively during summer months for larger particles, a phenomenon driven by accelerated aging processes and more homogeneous internal mixing. No comparable effects are detected in other seasons.

## 4. Summary and conclusions

This study examines size-resolved and seasonal variations in aerosol hygroscopicity and their influencing factors in northern Nanjing, based on field observations at NUIST in 2021 using H-TDMA measurements.





Aerosols in this suburban region exhibit lower hygroscopicity than in other areas, primarily due to higher organic content
(annual $MF_{OA}$ in $PM_{2.5}$: 42.92%). The $\kappa_{mean}$ increases with particle size across all seasons, especially for nucleation-mode
particles. Annual $\kappa_{mean}$ values (± standard deviations) for 40, 80, 110, 150, and 200 nm particles are 0.12 ± 0.04, 0.16 ± 0.05,
0.18 ± 0.05, 0.20 ± 0.05, and 0.22 ± 0.06, respectively.

Winter and spring show higher $\kappa_{mean}$ due to elevated levels of $MF_{SNA}$ and stronger $\kappa_{MH}$, driven by $PM_{2.5}$ pollution. In contrast,
summer and autumn exhibit lower $\kappa_{mean}$ and weaker size dependence. For nucleation-mode particles, $\kappa_{mean}$ varies significantly
by size in winter and spring but remains stable in other seasons. Accumulation-mode particles display weaker seasonal
variation. In winter, high $NO_3^-$ content boosts $\kappa_{MH}$, especially in accumulation-mode particles. However, $\kappa_{mean}$ is highest in
spring due to increased SNA content and $NF_{MH}$. Notably, 40 nm particles in spring show the highest $\kappa_{mean}$ due to both increased
$NF_{MH}$ and decreased $NF_{NH}$.

Across all seasons, $\kappa_{mean}$ is mainly governed by $NF_{MH}$. For nucleation-mode particles, $\kappa_{mean}$ is also influenced by $NF_{NH}$, $\kappa_{MH}$,
and $\kappa_{LH}$—particularly at 40 nm—while accumulation-mode particles are more affected by $\kappa_{MH}$. In summer and autumn, the
impact of $\kappa_{MH}$ diminishes, while $\kappa_{LH}$ becomes more influential. Photochemical processes increase $SO_4^{2-}$ and SOA during
daytime (except winter), while $NO_3^-$ accumulates at night via aqueous-phase reactions. As a result, 200 nm particles show
afternoon and evening $\kappa_{mean}$ peaks. Traffic emissions cause $\kappa_{mean}$ dips across all sizes during morning and evening rush hours.
Spring records the highest frequency of new particle formation (NPF) events (20.65%). During these events, newly formed
particles are likely organic-rich, leading to lower $\kappa_{mean}$ for 40 nm particles in the morning. As particles age, raising $\kappa_{mean}$ for
200 nm particles by late afternoon. Compared to winter, nucleation-mode particles on spring non-NPF days and spring non-
NPF days show much higher $\kappa_{mean}$, while accumulation-mode particles are similar on spring non-NPF days but significantly
lower than spring NPF days (except for 110 nm). Compared to spring non-NPF days, $\kappa_{mean}$ on spring NPF days decreases (by
6.76%) for 40 nm particles, and increases (by 5.07%) for 200 nm particles.

Intra-seasonal hygroscopicity variations are much smaller than inter-seasonal differences, largely due to the greater chemical
contrast among seasonal air masses. Spatial controls differ by particle size: nucleation-mode particles are primarily influenced
by local sources, as indicated by stable $\kappa_{mean}$ despite NF variability. In contrast, within a given season, accumulation-mode
particles transported from the east and north show enhanced hygroscopicity due to higher SNA levels. Notably, long-range air
mass transport enhances $\kappa_{mean}$ for larger particles only in summer, likely due to accelerated aging and more uniform internal
mixing—an effect not observed in other seasons.

These findings provide valuable insights into the complex interactions between aerosol chemical composition, particle size,
seasonal meteorological conditions, and regional air mass transport in shaping aerosol hygroscopicity. Understanding these
relationships is essential for improving the accuracy of regional climate models, particularly in estimating aerosol–cloud
interactions and radiative forcing. Moreover, the study highlights the critical role of local emissions and secondary processes
in influencing aerosol properties, offering a scientific basis for air quality management and pollution control strategies in
rapidly urbanizing regions like Nanjing and the broader Yangtze River Delta.





*Data availability.* Data used in the study are available at http://gofile.me/5JhP4/wZoKYiAJn.

*Author contributions.* YW designed the experiment; JZ, JX and YW carried it out and analysed the data. Other co-authors participated in science discussions and suggested analyses. JZ prepared the paper with contributions from all co-authors.

*Competing interests.* The authors declare that they have no conflict of interest.

*Acknowledgements.* This work is funded by National Natural Science Foundation of China (NSFC) research projects (Grant Nos. 42030606). We thank all participants in the field campaign for their tireless work and cooperation.



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
