# Peer review of "Measurement report: Size-resolved and seasonal variations in aerosol hygroscopicity dominated by organic formation and aging: Insights from a year-long observation in Nanjing"

_EGUsphere, 2025_

## Referee Comment (RC2)

In this manuscript, Zhang et al. investigate the aerosol hygroscopicity based on year-long measurement in an urban environment. Measurement data presented is sufficient for a measurement report, however, in its current state I cannot recommend the manuscript for publication.

[1] The authors consider all particles smaller than 100 nm as nucleation mode particles. 40 nm and 80 nm particles cannot be called newly formed particles. These were either emitted as primary particles or have been growing from newly formed particles for multiple hours. I would recommend using the common classification into nucleation, Aitken and accumulation mode, and modify the interpretation of the results throughout the entire manuscript accordingly.

[2] The authors state that an increased mass fraction of SNA in the larger sizes (80 – 200 nm) during winter and spring and more organics in the smaller sizes (40 – 80 nm) affect the hygroscopicity (lines198-199 and 206-208). Looking at the total organic mass in Fig. 1b the highest values can be observed during summer followed by autumn. One should be careful with these statements as no size-resolved composition was measured. In addition, ON and OS could be present. Is the measured $NO_3^-$ and $SO_4^{2-}$ plainly inorganic?

[3] This is related to comment [1]. The authors state that particles of 40 nm and 80 nm primarily originate from direct emissions or NPF events with limited aging (lines 250-251). These are not in the size range of newly formed particles and can have been aged. In addition, POA can also be observed in the accumulation mode for example from diesel engines.

[4] Which matrix was used to calculate the $R^2$. It would be helpful to mention which $R^2$ value is considered a good correlation, or to have an impact on the mean hygroscopicity. This includes for example the statement that the mean hygroscopicity for accumulation-mode particles is significantly affected by $\kappa_{MH}$ (lines 260-261).

[5] The authors state that traffic-induced compositional changes reduce mean hygroscopicity during rush hours (lines 271-272 and Fig. 5), and that the suppression effect is the greatest during summer (lines 272-273). I am not convinced about this statement. Several sizes and seasons show no decease in mean hygroscopicity during rush hours, but mean hygroscopicity does decrease for some sizes during non-summer months during rush hours. The observed increase in the particle number concentration in winter and spring during rush hours in Fig. 4 is also not reflected during summer.

[6] Lines 287-294: What is meant by gas-to-particle conversion through aqueous-phase oxidation processes? $NO_3$ radical chemistry? The $NO_3$ mass fraction in Fig. 4 does not show an inverse diurnal pattern during winter. Could you please elaborate on that? How was it concluded that only accumulation-mode particles are affected? The 200 nm particles in Fig. 5 do not show the bimodal distribution stated, nor does the $NF_{MH}$ increase.

[7] How were NPF days and events identified? Figure 4a2 does not clearly show a banana-shaped diurnal cycle indicating nucleation events (lines 295-297).

[8] I do not see the gradual decrease of $MF_{POA}$ in Fig. S1.

[9] What is meant with mean hygroscopicity increases slightly as the particles grow (lines 308-310)? There is no difference in non-NPF and NPF days for mean hygroscopicity of 40 nm particles.

[10] Not for all sizes a distinct diurnal peak is visible at 16:00 LT in Fig. S2 and Fig. S3. The same applies to $NF_{MH}$ (lines 312-313). I don't think the conclusion in lines 313-315 can be drawn from Fig. S2 and Fig. S3. The mean hygroscopicity does not change throughout the day for the different particle sizes for both non-NPF and NPF days.

[11] Which figure shows the results mentioned in lines 319-322?

[12] The conclusions in lines 335-341 are confusing. How can there be a size-dependent behaviour for one particle size?

All the results in section 3.2 should be re-interpreted taken comment [1] into account.

[13] Line 349-350: "...C2 exhibiting substantially lower concentrations than C1 and C3 (Fig. S4)." Concentrations of what? For a better understanding of Fig. S4 it would be nice to mention in the main manuscript that the total mass is represented by the size of the pie chart, not only in the caption. Even better would be to present the actual mass concentration in a table or in Fig. S4. I don't think the conclusion in lines 352-353 can be drawn just from the $PM_{2.5}$ mass concentration. Looking at summer for example C2 also shows lower mass concentrations compared to C1 and C3, but nucleation events occur only at frequency of around 3% according to Table S2.

[14] The influence of $NF_{NH}$ and $NF_{MH}$ during all seasons except autumn seem to be the same for 40 nm particles (line 380).

[15] Unfortunately, only the back trajectories for autumn were available, but looking at those all clusters show long-range transport. How does enhanced photochemical aging lead to more internal mixing (lines 388-390)?

[16] "...lower hygroscopicity than in other areas..." Which areas is it compared to?

[17] Which meteorological conditions are you referring to in line 436? Temperature, wind speed, wind direction, and RH for example are not mentioned in this manuscript. Local emission sources, besides traffic emissions were not linked to aerosol properties (lines 439).

Other comments

Line 30: …suspended in a gas… is the commonly used definition.

The HTDMA and CCNC measure in different saturation regimes. Thus, comparison of hygroscopicity measurements in the HTDMA and the CCNC is meaningless (lines 43-50). Please modify.

It would be nice to mention the ACSM in the introduction as it is used throughout the entire manuscript.

Some statements are vague or confusing, for example lines 32-33, lines 40-41, lines 104-105, lines 223-224, lines 241-242.

Line 212: 40 nm particles during winter and summer also show similar hygroscopicity in Fig. 1a

Line 230: "…possesses greater hygroscopicity." compared to what?

Line 231: "…more pronounced seasonal contrast." compared to what?

One should be careful with words like significantly or substantial (for example lines 231, 261, and 289). Please check throughout the manuscript.

---

## Author Response (AR1)

**Response to Reviewers**

Thanks for the reviewers' insightful comments on our manuscript entitled "*Measurement report: Size-resolved and seasonal variations in aerosol hygroscopicity dominated by organic formation and aging: Insights from a year-long observation in Nanjing*". We greatly appreciate the valuable feedback, which has been instrumental in improving the quality of our work. All comments have been carefully considered and addressed. In accordance with the instructions provided in your letter, we have updated the revised manuscript accordingly. Our responses are highlighted in blue, and changes made to the manuscript are marked in red for ease of reference. Please note that the line numbers referenced in our response correspond to those in the original manuscript submission, not the revised version with tracked changes.

**Responses to RC1**

This manuscript investigates aerosol hygroscopicity in an urban environment based on year-long measurements. The paper is not particularly novel but, the length of the data set combined with a careful scientific analysis makes the paper certainly worthy of publication. I have a few issues that should be addressed before that final acceptance.

**Comment 1:** The authors call particles smaller than 100 nm in diameter a nucleation mode. This is inconsistent with current scientific literature which usually separates ultrafine (<100 nm) particles into nucleation and Aitken modes, the border between these two being in the range 20-30 nm. Considering that the smallest particles considered in this work have a diameter of 40 nm, I strongly recommend that the authors call <100 nm particles as Aitken mode particles, not nucleation mode particles.

**Response:** We fully agree that the terminology should align with current scientific conventions. Accordingly, we have revised the manuscript throughout—including the abstract, main text, and figure captions—by uniformly updating references to particles smaller than 100 nm to "ultrafine particles" and those in the 40–100 nm size range to "Aitken mode". We have made the following adjustment to section 2.2.2: "Particles are classified into two modes based on their diameter: ultrafine-mode ($D_\mathrm{p} \leq 100$ nm) and accumulation mode (100 nm $< D_\mathrm{p} \leq 700$ nm). Specifically, particles within the 40 nm $< D_\mathrm{p} \leq 100$ nm

range are classified as belonging to the Aitken mode. Then the total number concentrations in ultrafine- and accumulation-mode particles ($N_{ult}$ and $N_{acc}$) are then calculated, separately. "

**Comment 2:** Related to the previous comment, the authors call 40 nm particles as newly formed nucleation mode particles. Such particle cannot really be called newly formed, not even recently formed, since for typical growth rates of newly formed (a few nm in diameter) particles, its takes in minimum a few hours to reach 40 nm. Again, I would recommend calling 40 nm particles something else than newly formed particles. For example, small particles originating from NPF (e.g. line 322) might be an appropriate term.

**Response:** We agree that "newly formed" is not accurate for 40 nm particles. We have replaced phrases such as "newly formed nucleation mode particles" with more precise descriptions, such as "Aitken-mode particles originating from new particle formation (NPF) events". This change better reflects the evolutionary state of these particles. We have made the following adjustment to lines 306–308: " Additionally, in contrast to spring non-NPF days, a slightly lower $\kappa_{mean}$ is observed for 40 nm particles (Fig. S2a1), suggesting that the Aitken-mode particles originating from NPF events exhibits lower hygroscopicity. " Lines 314:"These findings suggest that Aitken-mode particles originating from NPF events, which exhibit lower hygroscopicity..."

**Comment 3:** One should be careful when discussing size-dependent behavior of different quantities, as the natural measured of particle size scale logarithmically rather than linearly. For example, kappa parameter may appear having a larger size-dependency in a linear scale (Fig. 1a, lines 205-208), but this may not be the case when comparing the change in kappa from 40 to 80 nm to that from 100 nm to 200 nm (log diameter scale).

**Response:** We thank the reviewer for highlighting this important aspect of size distribution analysis. We separately plotted the relationship between $\kappa_{mean}$ and $D_p$, as well as between $\kappa_{mean}$ and $\log_{10}(D_p)$ (as shown in Fig. R1a and R1b below). We found that regardless of the plotting method, $\kappa_{mean}$ exhibits size dependency with generally consistent trends. Furthermore, previous studies have mostly presented linear plots of $\kappa_{mean}$ versus $D_p$, and while very few have reported logarithmic plots of $\kappa_{mean}$ versus $D_p$ (Hong et al., 2024; Peng et al., 2020; Ray et al., 2023; Shi et al., 2022; Wang et al., 2018). Therefore, we believe that linear plots of $\kappa_{mean}$ versus $D_p$ are sufficient to elucidate the relationship between $\kappa_{mean}$ and $D_p$, and

thus no comparison with logarithmic plots has been added in the revised manuscript. In addition, we found that when comparing the change in $\kappa_{mean}$ from 40 nm to 80 nm with that from 100 nm to 200 nm, the plots of these two different particle size scales in Fig. R1 show different rates of change in hygroscopicity with particle size. Therefore, to avoid controversial discussion results, we deleted the discussion on the rate of change of $\kappa_{mean}$ with particle size in the manuscript. For example, in section 3.1.1: "", "" In section 4: ""

[Figure]

Figure R1. (a) Variations in the mean hygroscopicity parameter ($\kappa_{mean}$) with particle diameter ($D_p$) across different seasons. (b) Variations in $\kappa_{mean}$ with particle log diameter ($\log_{10}(D_p)$) across different seasons. The shaded error bands represent the standard deviation of $\kappa_{mean}$.

**Comment 4:** The authors state that traffic emission reduce kappa across all particles sizes during rush hours (lines 271-272). I am not convinced about this statement: when looking at fig. 5, there are clearly

seasons and size ranges for which no clear decrease in kappa is visible during rush hours. Please modify the statement.

**Response:** We agree that our original statement was too absolute. A decrease during rush hours is not observed for all periods and particle sizes, such as the hygroscopicity of 110 nm and 150 nm particles in spring and autumn (Fig. R2). We have modified the sentence (Lines 271-272) to:" Traffic-related compositional changes during rush hours, especially increased $NF_{NH}$, systematically reduce $\kappa_{mean}$ across most particle size ranges, with magnitudes varying by particle size and season (Fig. 5). Compared to the Nanjing site in this study, sites like Madrid and Budapest show a more pronounced decline in $\kappa_{mean}$ during traffic emissions rush hours (Alonso-Blanco et al., 2019; Enroth et al., 2018). This disparity implies that aerosol hygroscopicity at the Nanjing site is less sensitive to rush-hour traffic emissions." This more nuanced description better reflects the data presented in Fig. 5.

[Figure]

Figure R2. Diurnal variations of the $\kappa_{mean}$ for different size particles (40–200 nm) during different seasons.

**Comment 5:** The decrease in kappa due to traffic is observed to be the largest in summer (lines 272-273). What might be the reason for this? Based on particle number size distributions (Fig. 4a), the influence of traffic during rush hours is not particularly strong in summer, and appears even weaker than in winter or spring.

**Response:** The decrease in $\kappa_{mean}$ due to traffic is observed to be the largest in summer (lines 272-273). This is likely due to the relatively low total particle number concentration ($N_{tot}$) in summer, with traffic emissions during morning and evening rush hours constituting a larger fraction of $N_{tot}$, hence causing a relatively more pronounced decrease in $\kappa_{mean}$. We have reanalyzed the data and revised the discussion in Section 3.3 (Lines 271-273) as follows:

" Relative to ultrafine-mode particles, the number concentration of accumulation-mode particles has a greater impact on $PM_{2.5}$ mass concentration. In contrast to other seasons, summer noon shows a more noticeable increasing trend in the number concentration of accumulation-mode particles, which to some extent weakens the diurnal variation amplitude of $PM_{2.5}$ mass concentration in summer, resulting in less pronounced changes in $PM_{2.5}$ mass concentration during summer rush hours (Fig. 4c3). Traffic-related compositional changes during rush hours, especially increased $NF_{NH}$, systematically reduce $\kappa_{mean}$ across most particle size ranges, with magnitudes varying by particle size and season (Fig. 5). Compared to the Nanjing site in this study, sites like Madrid and Budapest show a more pronounced decline in $\kappa_{mean}$ during traffic emissions rush hours (Alonso-Blanco et al., 2019; Enroth et al., 2018). This disparity implies that aerosol hygroscopicity at the Nanjing site is less sensitive to rush-hour traffic emissions. Notably, while $PM_{2.5}$ mass concentration increases are less significant during summer rush hours than in other seasons (Fig. 4c3), the reductions in $\kappa$ values across most particle size ranges are more pronounced. This observation likely stems from the relatively low total particle number concentration ($N_{tot}$) in summer, whereby traffic emissions during morning and evening rush hours represent a larger proportion of $N_{tot}$, thereby inducing a relatively pronounced decrease in $\kappa_{mean}$."

Additionally, regarding the issue of the less noticeable variation in PNSD during the summer traffic emission period in Fig. 4a3, this may be because all seasons share a common colorbar, resulting in visually subdued PNSD changes during the summer traffic emission period when number concentrations are relatively low. If the colorbar is readjusted, we would observe that the particle number concentration during the summer traffic emission period still shows a significant increase (see Fig. R3b3 below).

[Figure]

Figure R3. (a1-a4) When the limit of colorbar of $dN/d\log D_p$ is [0, 12000], the diurnal variation of particle

number size distribution (PNSD), total particle number concentration ($N_{tot}$), accumulation-mode particle number concentration ($N_{acc}$) and ultrafine-mode particle number concentration ($N_{ult}$). (b1–b4) When the limit of colorbar of $dN/dlogD_p$ is [0, 8000], the diurnal variation of PNSD, $N_{tot}$, $N_{acc}$ and $N_{ult}$.

**Comment 6:** The statement on lines 335-339 is strange: what is meant by a size dependency of 110 nm particles, as having a size dependency necessarily involves multiple sizes?

**Response:** We apologize for the unclear phrasing. The sentence has been rewritten for clarity (Lines 335-339). It now reads: " Particles with a diameter of 110 nm represent a transitional size between the Aitken and accumulation modes. Their hygroscopicity in winter is slightly higher than that observed during spring NPF days, but slightly lower than during spring non-NPF days."

**Comment 7:** Saying that the NPF event enhance the hygroscopicity of accumulation mode particles gives an impression of a cause-effect relationship between these two. I do not think such a conclusion can be drawn. It is at least equally possible that there are certain condition that simultaneously favor NPF and high hygroscopicity of accumulation mode particles.

**Response:** We agree with the reviewer that our wording implied causation where only correlation was observed. The revised manuscript emphasizes that hygroscopicity of accumulation-mode particles is enhanced on NPF event days, rather than suggesting that NPF events directly promote the enhancement of hygroscopicity in accumulation-mode particles.

We have made the following revisions in the manuscript: " These findings collectively demonstrate that, compared to spring non-NPF days, spring NPF days exhibit slightly reduced hygroscopicity in Aitken-mode particles and slightly enhanced hygroscopicity in accumulation-mode particles. Regardless of the influence of NPF events, particles of all sizes in spring demonstrate relatively stronger hygroscopicity than those of the same size in other seasons, except for 110 nm particles, which show slightly lower hygroscopicity during spring NPF days compared to winter." Furthermore, we have removed the ambiguous statement: ""

**Comment 8:** line 409: saying that lower than in other areas gives an impression that this location has the lowest kappa of all urban areas. I suppose this is not what the authors mean. Please modify.

**Response:** Thank you for catching this overgeneralization. We have modified the sentence (Line 409) to: "Aerosols in this suburban region exhibit relatively low hygroscopicity, which may be mainly due to relatively higher organic content (annual average $MF_{OA}$ in $PM_{2.5}$: 42.92%)."

**Response to Minor/Technical Issues**

**Comment 9:** line 231: more pronounced compared to what? Compared to Aitken or ultrafine particles?

**Response:** We have clarified the sentence (Line 231) to specify the comparison: "In general, aerosol particles exhibit higher $\kappa_{MH}$ in winter than in spring, which may be attributed to the higher abundance of $NO_3^-$ in winter aerosols leading to enhanced hygroscopicity of MH groups. This effect is more pronounced in accumulation-mode particles compared to Aitken-mode particles, with $\kappa_{MH}$ showing more distinct seasonal variations in the former…"

**Comment 10:** line 296: The stated accuracy (20.65%) is overly high. 21% would be more reasonable. Please check out throughout the paper.

**Response:** We have corrected the number to 21% throughout the manuscript (Line 296 and elsewhere). Furthermore, we have modified the sentence (Line 351) to: " As indicated in Table S4, the occurrence frequencies of NPF events during spring for C1, C2, and C3 are 11%, 42%, and 25%, respectively…"

**Comment 11:** line 331: similar to …

**Response:** We have corrected the typo to "Similar to spring, …"

**Comment 12:** line 416: … a high … content …

**Response:** We have corrected the grammar to "In winter, a high $NO_3^-$ content boosts $\kappa_{MH}$, …"

**Comment 13:** lines 425-426: The sentence "As particles … by late afternoon" does not work in its present form. Please modify.

**Response:** We have rewritten the sentence for clarity (Lines 425-426). It now reads: " With the aging of particles, the $\kappa_{\mathrm{mean}}$ for 200 nm particles slightly enhanced in the late afternoon. "

**References:**

Alonso-Blanco, E., Gómez-Moreno, F.J., and Artíñano, B.: Size-resolved hygroscopicity of ambient submicron particles in a suburban atmosphere, Atmos Environ, 213, 349-358, https://doi.org/10.1016/j.atmosenv.2019.05.065, 2019.

Enroth, J., Mikkilä, J., Németh, Z., Kulmala, M., and Salma, I.: Wintertime hygroscopicity and volatility of ambient urban aerosol particles, Atmos Chem Phys, 18, 4533-4548, 10.5194/acp-18-4533-2018, 2018.

Hong, J., Ma, J., Ma, N., Shi, J., Xu, W., Zhang, G., Zhu, S., Zhang, S., Tang, M., Pan, X., Xie, L., Li, G., Kuhn, U., Yan, C., Qi, X., Zha, Q., Nie, W., Tao, J., He, Y., Zhou, Y., Sun, Y., Xu, H., Liu, L., Cai, R., Zhou, G., Kuang, Y., Yuan, B., Wang, Q., Petäjä, T., Kerminen, V., Kulmala, M., Cheng, Y., and Su, H.: Low Hygroscopicity of Newly Formed Particles on the North China Plain and Its Implications for Nanoparticle Growth, Geophys Res Lett, 51, e2023GL107516, https://doi.org/10.1029/2023GL107516, 2024.

Peng, C., Wang, Y., Wu, Z., Chen, L., Huang, R.J., Wang, W., Wang, Z., Hu, W., Zhang, G., Ge, M., Hu, M., Wang, X., and Tang, M.: Tropospheric aerosol hygroscopicity in China, Atmos Chem Phys, 20, 13877-13903, https://doi.org/10.5194/acp-20-13877-2020, 2020.

Ray, A., Pandithurai, G., Mukherjee, S., Kumar, V.A., Hazra, A., Patil, R.D., and Waghmare, V.: Seasonal variability in size-resolved hygroscopicity of sub-micron aerosols over the Western Ghats, India: Closure and parameterization, Sci Total Environ, 869, 161753, https://doi.org/10.1016/j.scitotenv.2023.161753, 2023.

Shi, J., Hong, J., Ma, N., Luo, Q., He, Y., Xu, H., Tan, H., Wang, Q., Tao, J., Zhou, Y., Han, S., Peng, L., Xie, L., Zhou, G., Xu, W., Sun, Y., Cheng, Y., and Su, H.: Measurement report: On the difference in aerosol hygroscopicity between high and low relative humidity conditions in the North China Plain, Atmos Chem Phys, 22, 4599-4613, https://doi.org/10.5194/acp-22-4599-2022, 2022.

Wang, Y., Li, Z., Zhang, Y., Du, W., Zhang, F., Tan, H., Xu, H., Fan, T., Jin, X., Fan, X., Dong, Z., Wang, Q., and Sun, Y.: Characterization of aerosol hygroscopicity, mixing state, and CCN activity at a suburban site in the central North China Plain, Atmos Chem Phys, 18, 11739-11752, https://doi.org/10.5194/acp-18-11739-2018, 2018.

**Responses to RC2**

In this manuscript, Zhang et al. investigate the aerosol hygroscopicity based on year-long measurement in an urban environment. Measurement data presented is sufficient for a measurement report, however, in its current state I cannot recommend the manuscript for publication.

**Comment [1]:** The authors consider all particles smaller than 100 nm as nucleation mode particles. 40 nm and 80 nm particles cannot be called newly formed particles. These were either emitted as primary particles or have been growing from newly formed particles for multiple hours. I would recommend using the common classification into nucleation, Aitken and accumulation mode, and modify the interpretation of the results throughout the entire manuscript accordingly.

**Response:** We fully agree that the terminology should align with current scientific conventions. Accordingly, we have revised the manuscript throughout—including the abstract, main text, and figure captions—by uniformly updating references to particles smaller than 100 nm to "ultrafine particles" and those in the 40–100 nm size range to "Aitken mode". Furthermore, we agree that "40 nm and 80 nm particles cannot be called newly formed particles". We have replaced phrases such as "newly formed nucleation mode particles" with more precise descriptions, such as "Aitken-mode particles originating from new particle formation (NPF) events". This change better reflects the evolutionary state of these particles. We have made the following adjustment to lines 167–169: "Particles are classified into two modes based on their diameter: ultrafine-mode ($D_p \leq 100$ nm) and accumulation mode (100 nm < $D_p \leq$ 700 nm). Specifically, particles within the 40 nm < $D_p \leq 100$ nm range are classified as belonging to the Aitken mode. Then the total number concentrations in ultrafine- and accumulation-mode particles ($N_{ult}$ and $N_{acc}$) are then calculated, separately. "

**Comment [2]:** The authors state that an increased mass fraction of SNA in the larger sizes (80 – 200 nm) during winter and spring and more organics in the smaller sizes (40 – 80 nm) affect the hygroscopicity (lines198-199 and 206-208). Looking at the total organic mass in Fig. 1b the highest values can be observed during summer followed by autumn. One should be careful with these statements as no size-resolved composition was measured. In addition, ON and OS could be present. Is the measured $NO_3^-$ and $SO_4^{2-}$ plainly inorganic?

**Response:** We thank the reviewer for this insightful comment and agree that caution is warranted when inferring size-dependent composition without direct measurements. Below, we address each aspect of the query, including the basis for our statements, the reconciliation with bulk organic aerosol seasonality in Fig. 1b, and the potential role of organic nitrates (ON) and organosulfates (OS).

We did not directly measure size-resolved chemical composition, particularly for Aitken-mode particles. As such, our statements about an increased mass fraction of secondary sulfate–nitrate–ammonium (SNA) in larger sizes (80–200 nm) during winter and spring, and more organics in smaller sizes (40–80 nm), are presented as plausible mechanisms rather than definitive attributions. This is already indicated in the manuscript (e.g., lines 206–208: "This is likely due to the higher organic content..."). These inferences are supported by prior studies in Nanjing, where Wang et al. (2022) reported that Aitken-mode particles contain a larger proportion of organic matter and black carbon, with less inorganic matter. Additionally, bulk aerosol composition from the Aerosol Chemical Speciation Monitor (ACSM) and Aethalometer can reasonably reflect variations in accumulation-mode particles, as this mode dominates fine-mode mass (Wang et al., 2019). Our approach aligns with Yuan et al. (2020), who integrated $PM_{2.5}$ and H-TDMA data to investigate hygroscopicity without size-resolved composition.

We acknowledge that Fig. 1b shows the highest bulk organic mass in summer, followed by autumn. Our discussion focuses on relative size-dependent compositional influences (e.g., a higher organic fraction in the 40–80 nm range) as a potential driver of hygroscopicity ($\kappa$) patterns, rather than absolute bulk organic concentrations. This is consistent with seasonal source variations and prior regional observations, but we emphasize it as a hypothesis grounded in indirect evidence.

The ACSM effectively separates organic and inorganic aerosol components in $PM_{2.5}$, with $NO_3^-$ and $SO_4^{2-}$ signals predominantly reflecting inorganic nitrate and sulfate under typical urban conditions. However, we recognize that unit-mass resolution ACSM cannot fully distinguish minor contributions from ON or OS without high-resolution data. Studies in Nanjing indicate that inorganic ions (ammonium, sulfate, nitrate) dominate $PM_{2.5}$ mass, followed by organic matter and elemental carbon (Liu et al. 2021), and that ON constitutes a very minor component of OA in summer (e.g., 0.6-1.1% based on Xian et al. 2023). Thus, ON and OS are likely a relatively small fraction in our dataset. Even if present, both are secondary products and generally more hygroscopic than primary OM (Aklilu et al., 2006), meaning any unaccounted ON/OS would conservatively underestimate the $\kappa$ suppression attributed to less hygroscopic organics. Therefore, we believe ON/OS have minimal impact on our conclusions.

To enhance clarity and address the reviewer's concerns, we propose the following manuscript revisions:

Revise lines 198–199 and 206–208 to explicitly frame the interpretations as hypothetical:

Original (lines 198–199): "... This pattern may be attributed to the increased mass fractions of more hygroscopic sulfate–nitrate–ammonium (SNA) during the more severe $PM_{2.5}$ pollution in winter and spring...."

Revised: "...This pattern may be attributed to the increased mass fractions of hydrophilic sulfate–nitrate–ammonium (SNA) during the more severe $PM_{2.5}$ pollution in winter and spring..."As size-resolved composition was not measured, we present this as a plausible mechanism supported by regional studies (e.g., Wang et al., 2022) rather than a direct attribution.

The discussion on the $\kappa_{\mathrm{mean}}$ increase rate with particle size has some controversial issues in the manuscript. Therefore, we deleted the discussion on the rate of change of $\kappa_{\mathrm{mean}}$ with particle size, including the original (lines 206–208): "...This is likely due to the higher organic content..."

**Comment [3]:** This is related to comment [1]. The authors state that particles of 40 nm and 80 nm primarily originate from direct emissions or NPF events with limited aging (lines 250-251). These are not in the size range of newly formed particles and can have been aged. In addition, POA can also be observed in the accumulation mode for example from diesel engines.

**Response:** We appreciate the reviewer's comment and its connection to Comment [1], and we agree that particles in the 40–100 nm range can undergo some aging and that POA may appear in the accumulation mode, such as from diesel emissions. To address this, we have revised the manuscript for consistency by referring to particles in the 40–100 nm size range as "Aitken-mode particles." The updated description— " This phenomenon may be attributed to the fact that Aitken-mode particles, primarily originating from direct emissions or NPF events with limited aging "—clarifies that these particles result from NPF processes rather than being directly generated by NPF, and it acknowledges the potential for limited atmospheric processing.

This interpretation is supported by regional and comparative studies. For instance, Wang et al. (2022) found that in Nanjing, Aitken-mode particles contain a larger proportion of organic matter and black carbon compared to accumulation-mode particles, which show decreased organic fractions due to higher inorganic content. Similarly, Gysel et al. (2007) and Li et al. (2023) reported at other sites that Aitkenmode particles have a higher organic matter fraction relative to accumulation-mode particles. While our manuscript highlights that sub-100 nm particles may contain more weakly hygroscopic components such as organic aerosol, we do not exclude the presence of traffic-derived POA in accumulation-mode particles. Instead, we emphasize relative differences: accumulation-mode particles are also influenced by traffic emissions but typically contain lower proportions of weakly hygroscopic components like POA and black carbon compared to sub-100 nm particles. This framing aligns with the observed hygroscopicity patterns and does not imply that accumulation-mode particles are free of POA.

**Comment [4]:** Which matrix was used to calculate the R². It would be helpful to mention which R² value is considered a good correlation, or to have an impact on the mean hygroscopicity. This includes for example the statement that the mean hygroscopicity for accumulation mode particles is significantly affected by $\kappa_{MH}$ (lines 260-261).

**Response:** We have added the following content in the appendix as a supplement to the formula for calculating $R^2$:

"Supplementary Text 1: Calculation of correlation coefficient $R^2$.

The $R^2$ value is calculated using the following formula:

$$r = \frac{\sum_{i=1}^{n}(x_i - \bar{x})(y_i - \bar{y})}{\sqrt{\sum_{i=1}^{n}(x_i - \bar{x})^2 \cdot \sum_{i=1}^{n}(y_i - \bar{y})^2}} , \tag{1}$$

$$R^2 = r^2 , \tag{2}$$

where $r$ represents pearson correlation coefficient, $R^2$ represents correlation coefficient, $x_i$ and $y_i$ represent the observed values of the two variables, $\bar{x}$ and $\bar{y}$ denote their respective means, and $n$ is the sample size (Wang et al., 2020). "

In the manuscript, the discussion of correlations primarily focuses on comparing the relative magnitudes of $R^2$ between $\kappa_{mean}$ and various parameters (e.g., $\kappa_{MH}$, $\kappa_{NH}$, $\kappa_{LH}$) to assess their influence on mean hygroscopicity. A higher $R^2$ value indicates a stronger relationship and thus greater impact. For example, the statement that "$\kappa_{mean}$ for accumulation-mode particles is significantly affected by $\kappa_{MH}$" (lines 260–261) is made in comparison to lower $R^2$ values for other parameters, such as $\kappa_{NH}$ and $\kappa_{LH}$. We use $R^2 = 0.5$ as a general threshold for distinguishing meaningful correlations, where bluer colors in Fig. 3 indicate

poorer correlations ($R^2 < 0.5$) and yellower colors indicate stronger ones ($R^2 \geq 0.5$). This color-coded visualization helps highlight which factors most substantially drive variations in $\kappa_{mean}$.

**Comment [5]:** The authors state that traffic-induced compositional changes reduce mean hygroscopicity during rush hours (lines 271-272 and Fig. 5), and that the suppression effect is the greatest during summer (lines 272-273). I am not convinced about this statement. Several sizes and seasons show no decease in mean hygroscopicity during rush hours, but mean hygroscopicity does decrease for some sizes during non-summer months during rush hours. The observed increase in the particle number concentration in winter and spring during rush hours in Fig. 4 is also not reflected during summer.

**Response:** A decrease during rush hours is not observed for all periods and particle sizes, such as the hygroscopicity of 110 nm and 150 nm particles in spring and autumn (Fig. R2).

" Relative to ultrafine-mode particles, the number concentration of accumulation-mode particles has a greater impact on PM$_{2.5}$ mass concentration. In contrast to other seasons, summer noon shows a more noticeable increasing trend in the number concentration of accumulation-mode particles, which to some extent weakens the diurnal variation amplitude of PM$_{2.5}$ mass concentration in summer, resulting in less pronounced changes in PM$_{2.5}$ mass concentration during summer rush hours (Fig. 4c3). Traffic-related compositional changes during rush hours, especially increased NF$_{NH}$, systematically reduce $\kappa_{mean}$ across most particle size ranges, with magnitudes varying by particle size and season (Fig. 5). Compared to the Nanjing site in this study, sites like Madrid and Budapest show a more pronounced decline in $\kappa_{mean}$ during traffic emissions rush hours (Alonso-Blanco et al., 2019; Enroth et al., 2018). This disparity implies that aerosol hygroscopicity at the Nanjing site is less sensitive to rush-hour traffic emissions. Notably, while PM$_{2.5}$ mass concentration increases are less significant during summer rush hours than in other seasons (Fig. 4c3), the reductions in $\kappa$ values across most particle size ranges are more pronounced…" This more nuanced description better reflects the data presented in Fig. 5.

Additionally, regarding the issue of the less noticeable variation in PNSD during the summer traffic emission period in Fig. 4a3, this may be because all seasons share a common colorbar, resulting in visually subdued PNSD changes during the summer traffic emission period when number concentrations are relatively low. If the colorbar is readjusted, we would observe that the particle number concentration during the summer traffic emission period still shows a significant increase (see Fig. R3b3 below).

**Comment [6]:** Lines 287-294: What is meant by gas-to-particle conversion through aqueous-phase oxidation processes? $NO_3$ radical chemistry? The $NO_3$ mass fraction in Fig. 4 does not show an inverse diurnal pattern during winter. Could you please elaborate on that? How was it concluded that only accumulation-mode particles are affected? The 200 nm particles in Fig. 5 do not show the bimodal distribution stated, nor does the $NF_{MH}$ increase.

**Response:** We thank the reviewer for highlighting these points and apologize for any unclear phrasing in the original manuscript. We agree that elaboration on the nocturnal processes, diurnal patterns, and size-specific inferences is warranted, and we address each aspect below.

The phrase "gas-to-particle conversion through aqueous-phase oxidation processes" refers to the conversion of gaseous precursors (e.g., nitrogen oxides, NOx) into particulate nitrate ($NO_3^-$) via aqueous-phase reactions and gas-particle partitioning, facilitated by lower nocturnal temperatures and higher relative humidity (Sun et al., 2013). This can involve $NO_3$ radical chemistry, particularly at night when photolysis is absent, leading to enhanced nitrate formation in aqueous droplets. Regarding the diurnal pattern of $NO_3^-$ mass fraction in Fig. 4, we acknowledge that it shows an inverse cycle (nighttime maxima) during non-winter periods but not in winter, where levels remain consistently high throughout the day. This winter attenuation is likely due to stagnant meteorological conditions that limit pollutant dispersion and low temperature that inhibits nitrate decomposition, maintaining elevated $NO_3^-$ concentrations without strong diurnal variability.

On the size-specific effects, we lack direct size-resolved chemical composition data, so we cannot definitively assess compositional variations in Aitken-mode particles. Instead, our inferences about hygroscopicity influences are limited to accumulation-mode particles (e.g., 200 nm), which are closer in size to $PM_{2.5}$ and thus better reflected by bulk ACSM measurements (Wang et al., 2019; Yuan et al., 2020). For these particles, Fig. 5 illustrates two chemically-distinct diurnal patterns in $\kappa_{mean}$ during non-winter periods: an afternoon enhancement potentially driven by photochemical $SO_4^{2-}$ and SOA production, and a nighttime enhancement linked to $NO_3^-$ accumulation. We note that while the $NF_{MH}$ (more hygroscopic fraction) does not show a clear isolated increase, the overall $\kappa_{mean}$ enhancement arises from combined variations in NF across hygroscopicity groups ($NF_{MH}$, $NF_{LH}$, $NF_{NH}$). The nighttime hygroscopicity enhancement pattern is less evident in winter due to the attenuated $NO_3^-$ diurnal cycle, and we emphasize that this is an inferential interpretation based on bulk data.

To improve clarity and incorporate the reviewer's feedback, we have revised the manuscript as follows:

Revise lines 287–289: Original: "…This behavior is mechanistically explained through temperature mediated phase partitioning theory, where lower nocturnal temperatures coupled with higher relative humidity facilitate efficient gas-to-particle conversion through aqueous-phase oxidation processes (Sun et al., 2013)… " Revised: "This behavior may stem from lower nocturnal temperatures and higher RH, which promote efficient aqueous-phase reactions and gas-particle partitioning, potentially involving $NO_3$ radical chemistry, converting gaseous precursors like $NO_x$ into particulate $NO_3^-$ (Sun et al., 2013)."

Revise lines 290–294 to clarify two chemically-distinct diurnal patterns and winter exception: Original: "These compositional shifts drive distinct hygroscopicity dynamics in accumulation-mode particles. As illustrated in Fig. 5, the $\kappa_{mean}$ for 200 nm particles displays a bimodal diurnal pattern due to increased $NF_{MH}$... associated with $NO_3^-$ accumulation under favorable nighttime chemical conditions...." Revised: "These compositional shifts drive distinct hygroscopicity dynamics primarily in accumulation-mode particles (e.g., 200 nm), as their composition aligns more closely with bulk $PM_{2.5}$ measurements (Wang et al., 2019). As shown in Fig. 5, during non-winter periods, variations in NF among hygroscopicity groups result in two chemically-distinct diurnal patterns for $\kappa_{mean}$ in 200 nm particles: an afternoon enhancement (14:00–18:00 LT), likely driven by photochemical production of hydrophilic $SO_4^{2-}$ and SOA, and a nighttime enhancement (post-20:00 LT), associated with $NO_3^-$ accumulation under favorable nocturnal conditions. This pattern is less evident in winter, where stagnant meteorological conditions suppress pollutant dispersion, leading to consistently high $NO_3^-$ levels and minimal diurnal variation in $NO_3^-$ mass fraction (Fig. 4)."

**Comment [7]:** How were NPF days and events identified? Figure 4a2 does not clearly show a banana shaped diurnal cycle indicating nucleation events (lines 295-297).

**Response:** We thank the reviewer for this valuable question about our NPF identification methodology and the interpretation of Fig. 4a2, and we agree that the banana-shaped pattern in the seasonal average may appear subtle, which merits further clarification.

New particle formation (NPF) events involve the evolutionary process where newly formed particles with diameters smaller than 3 nm grow to larger sizes. Given the measurement limitations of our

instruments, SMPS (15 – 700 nm) and nano-SMPS (2 – 60 nm), we identify NPF events based on two primary criteria: (1) a significant increase in particle number concentration around 20 nm, and (2) a subsequent banana-shaped pattern in the particle number size distribution (PNSD) during the growth phase. Days meeting both criteria are classified as NPF event days. For instance, the PNSD on March 24, 2021 (spring) satisfied these conditions and was identified as a NPF event day (see Fig. R4 below). Due to frequent data gaps in nano-SMPS measurements, we did not include nano-SMPS-based diurnal PNSD variations in the manuscript. These data served only as an auxiliary tool for NPF event confirmation, with SMPS data as the primary basis.

The banana-shaped pattern in manuscript Fig. 4a2 is not highly distinct in the seasonal average because NPF events occurred on only 21% of spring days in our dataset, diluting the signal across all days. Nevertheless, as shown in Fig. S1a1, there is a notable increase in particle number concentration around the 20 nm size bin near 10:00 LT, with the size spectrum displaying a banana-shaped growth pattern from smaller to larger sizes. These features justify our classification of spring days into NPF and non-NPF categories for analysis.

[Figure]

Figure R4. (a) Particle Number Size Distribution (PNSD) time series recorded by SMPS on March 24, 2021. (b) PNSD time series recorded by Nano-SMPS on March 24, 2021.

To enhance clarity and address the reviewer's observation, we have revised lines 295 – 297 as follows: Original: " Figure 4a2 indicates that spring exhibits distinctive particle dynamics, characterized by frequent NPF events. The PNSD pattern displays a unique banana-shaped diurnal cycle, with the frequency of NPF occurrences reaching 20.65% during spring—approximately double the annual average and significantly higher than in other seasons (Table S2) …" Revised: "Figure 4a2 shows that

the spring afternoon period exhibits unique particle dynamics, with the size distribution displaying a subtle banana-shaped diurnal pattern (more evident in event-specific examples like Fig. S1a1). This pattern is potentially linked to the frequent NPF events that occur in spring. The frequency of NPF occurrences reaching 21% during spring (approximately double the annual average), significantly higher than in other seasons (Table S2)."

Additionally, to improve methodological transparency, we propose adding the following to the Methods section (e.g., Sect. 2.2.2): "NPF events are identified using SMPS data based on a significant particle number increase around 20 nm followed by banana-shaped PNSD growth, with nano-SMPS data being used auxiliarily for confirmation despite frequent gaps."

**Comment [8]:** I do not see the gradual decrease of $MF_{POA}$ in Fig. S1.

**Response:** Our original intent was to highlight that the proportion of POA within total OA decreases relative to SOA during spring NPF days, reflecting photochemical influences. However, as the reviewer notes, this trend is not distinctly demonstrated in Fig. S1. To address this and improve accuracy, we have revised lines 304–305 as follows: "$MF_{OA}$ significantly increases in spring NPF days, with $MF_{SOA}$ gradually increases over time…"

**Comment [9]:** What is meant with mean hygroscopicity increases slightly as the particles grow (lines 308-310)? There is no difference in non-NPF and NPF days for mean hygroscopicity of 40 nm particles.

**Response:** The statement "mean hygroscopicity increases slightly as the particles grow" refers to a modest increase in $\kappa_{mean}$ for Aitken-mode particles (e.g., 40–80 nm) from 08:00 to 12:00 LT during spring NPF days, as shown in Fig. R5a1–a2. While the overall difference in $\kappa_{mean}$ for 40 nm particles between non-NPF and NPF days is indeed small, this can be attributed to the generally low $\kappa_{mean}$ values for these smaller particles under most conditions, which exhibit less variability compared to accumulation-mode particles. As illustrated in Fig. R5a1, $\kappa_{mean}$ for 40 nm particles on NPF days shows a more pronounced decrease during morning and evening rush hours relative to non-NPF days, with values remaining slightly lower across most of the diurnal cycle.

To better align the manuscript with these observations and address the reviewer's point, we have revised lines 308–310 as follows: "As Aitken-mode particles grow during spring NPF days, the $\kappa_{mean}$ of 40 nm particles shows a modest increase from 08:00 to 12:00 LT, while the $\kappa_{mean}$ of 80 nm particles' increase is

relatively lagged from 10:00 to 14:00 LT (Fig. S2a1), though differences in $\kappa_{\text{mean}}$ for 40 nm particles between NPF and non-NPF days are minimal due to their generally low and less variable values."

[Figure]

Figure R5. Diurnal variations of the (a1–a5) $\kappa_{\text{mean}}$ for different size particles (40–200 nm) in spring NPF days (solid lines with dots) and spring non-NPF days (dashed lines with dots).

**Comment [10]:** Not for all sizes a distinct diurnal peak is visible at 16:00 LT in Fig. S2 and Fig. S3. The same applies to NF$_{\text{MH}}$ (lines 312-313). I don't think the conclusion in lines 313-315 can be drawn from Fig. S2 and Fig. S3. The mean hygroscopicity does not change throughout the day for the different particle sizes for both non-NPF and NPF days.

**Response:** We agree with the observation that a distinct diurnal peak at 16:00 LT is not a universal phenomenon across all particle sizes in Figs. S2 and S3. Therefore, we have made the following adjustment to lines 312-313: " A slightly increasing trend in $\kappa_{\text{mean}}$ occurs around 16:00 LT for most size particles, driven by concurrent increases in both the NF$_{\text{MH}}$ and $\kappa_{\text{MH}}$ (Figs. S2 and S3)." However, we argue that the mean hygroscopicity of particles of different sizes does exhibit diurnal variations on both non-NPF and NPF days. The apparent lack of significant variation may be attributed to the use of a

common y-axis for the diurnal profiles of $\kappa_{mean}$ across all particle sizes. Figure R5 shows the adjusted diurnal variation plots of mean hygroscopicity for different particle sizes without a shared y-axis.

**Comment [11]:** Which figure shows the results mentioned in lines 319-322?

**Response:** The lines 319-322 conclusion is derived from Fig. S3 a1–a4. We have added the following statement: " This indicates that the NPF event may only have a relatively pronounced effect on the internal mixing state of 40 nm particles. "

**Comment [12]:** The conclusions in lines 335-341 are confusing. How can there be a size-dependent behaviour for one particle size? All the results in section 3.2 should be re-interpreted taken comment [1] into account.

**Response:** We apologize for the unclear phrasing. The sentence has been rewritten for clarity (Lines 335-341). It now reads: " Particles with a diameter of 110 nm represent a transitional size between the Aitken and accumulation modes. Their hygroscopicity in winter is slightly higher than that observed during spring NPF days, but slightly lower than during spring non-NPF days."

We reinterpreted all the results in Section 3.2 in conjunction with comment [1]. Such as the following series of revisions in the manuscript:

Lines 295- 297: "Figure 4a2 shows that the spring afternoon period exhibits unique particle dynamics, with the size distribution displaying a subtle banana-shaped diurnal pattern (more evident in event-specific examples like Fig. S1a1). This pattern is potentially linked to the frequent NPF events that occur in spring. The frequency of NPF occurrences reaching 21% during spring (approximately double the annual average), significantly higher than in other seasons (Table S2)."

Lines 306-308: "This suggests that spring NPF events may be predominantly driven by the formation of SOA. Additionally, in contrast to spring non-NPF days, a slightly lower $\kappa_{mean}$ is observed for 40 nm particles (Fig. S2a1), suggesting that the Aitken-mode particles originating from NPF events exhibits lower hygroscopicity."

Lines 324-325: "These findings suggest that, compared to non-NPF days, the internal mixing of relatively

smaller particles is slightly weakened on NPF days, while relatively larger particles exhibit a slightly enhanced state of internal mixing."

Lines 335-341: "Particles with a diameter of 110 nm represent a transitional size between the Aitken and accumulation modes. Their hygroscopicity in winter is slightly higher than that observed during spring NPF days, but slightly lower than during spring non-NPF days. These findings collectively demonstrate that, compared to spring non-NPF days, spring NPF days exhibit slightly reduced hygroscopicity in Aitken-mode particles and slightly enhanced hygroscopicity in accumulation-mode particles. Regardless of the influence of NPF events, particles of all sizes in spring demonstrate relatively stronger hygroscopicity than those of the same size in other seasons, except for 110 nm particles, which show slightly lower hygroscopicity during spring NPF days compared to winter."

**Comment [13]:** Line 349-350: "…C2 exhibiting substantially lower concentrations than C1 and C3 (Fig. S4)." Concentrations of what? For a better understanding of Fig. S4 it would be nice to mention in the main manuscript that the total mass is represented by the size of the pie chart, not only in the caption. Even better would be to present the actual mass concentration in a table or in Fig. S4. I don't think the conclusion in lines 352-353 can be drawn just from the $PM_{2.5}$ mass concentration. Looking at summer for example C2 also shows lower mass concentrations compared to C1 and C3, but nucleation events occur only at frequency of around 3% according to Table S2.

**Response:** We have revised Fig. S4 in the manuscript (see Fig. R6), adding the display of actual $PM_{2.5}$ mass concentrations below the corresponding pie charts in the figure.

Regarding the conclusion in lines 352–353, it primarily explains that NPF events in Nanjing spring are more likely to occur during relatively clean periods, but does not imply that clean periods necessarily lead to NPF events. This conclusion represents a necessary but not sufficient condition. The occurrence of NPF events is not only dependent on atmospheric conditions with low pollution levels (low condensation sink, CS), but also related to gaseous precursors and meteorological conditions (Shang et al., 2023; Shen et al., 2023). Since data on gaseous precursors and meteorological conditions are not available in the manuscript, it is not possible to provide a more comprehensive explanation of the sufficient conditions that make cluster C2 more conducive to NPF occurrence in spring air masses. Therefore, we have made the following adjustment to lines 349–352:" As illustrated in Fig. S4, the size

of the pie charts represents the PM₂.₅ mass concentration in different clusters. Notably, springtime PM$_{2.5}$

mass concentrations show significant variation among air mass categories, with C2 exhibiting relatively

lower mass concentrations than C1 and C3. As indicated in Table S4, the occurrence frequencies of NPF

events during spring for C1, C2, and C3 are 11%, 42%, and 25%, respectively, suggesting that NPF

events in Nanjing spring are more likely under the cleaner conditions of C2, characterized by lower PM$_{2.5}$

mass concentrations. This pattern may stem from reduced condensation sinks and higher abundances of

gaseous precursors favorable for NPF in northerly air masses (Gysel et al., 2007; Li et al., 2023)."

[Figure]

Figure R6. The mass fractions (MF) of aerosol chemical components under the influence of different air

masses in different seasons. The size of the pie charts represents the PM$_{2.5}$ mass concentration, with the

value of PM$_{2.5}$ mass concentration displayed below the corresponding pie chart. The border colors of the

pie charts indicate the air mass types, i.e., red for Cluster C1, green for Cluster C2, and blue for Cluster

C3. The background shading colors represent the seasons, i.e., blue for winter, green for spring, red for

summer, orange for autumn.

**Comment [14]:** The influence of $NF_{NH}$ and $NF_{MH}$ during all seasons except autumn seem to be the same for 40 nm particles (line 380).

**Response:** Upon reanalysis of the data, we found that $NF_{NH}$ and $NF_{MH}$ exert comparable influences on $\kappa_{mean}$ for 40 nm particles, contrary to the statement in lines 380 that "$\kappa_{mean}$ is predominantly influenced by $NF_{NH}$ rather than $NF_{MH}$." Therefore, we have made the following revisions in the manuscript:

Line 248-250: "For Aitken-mode particles, $\kappa_{mean}$ is also affected largely by $NF_{NH}$, particularly for 40 nm particles. In autumn, the $R^2$ between $NF_{NH}$ and $\kappa_{mean}$ ($R^2 = 0.77$) for 40 nm particles is even slightly higher than that between $NF_{MH}$ and $\kappa_{mean}$ ($R^2 = 0.71$). "

Line 253: "These particles are characterized by higher $NF_{NH}$ and lower $NF_{MH}$, leading in an amplified influence of $NF_{NH}$ on the $\kappa_{mean}$ of Aitken-mode particles. "

Line 380: " For 40 nm particles, $\kappa_{mean}$ is predominantly influenced by $NF_{NH}$ and $NF_{MH}$ (Fig. 3), with $NF_{NH}$ being relatively higher and $NF_{MH}$ lower, particularly in autumn."

**Comment [15]:** Unfortunately, only the back trajectories for autumn were available, but looking at those all clusters show long-range transport. How does enhanced photochemical aging lead to more internal mixing (lines 388-390)?

**Response:** During our recent review of this manuscript, we identified an error in the presentation of Fig. 7 in the uploaded version, where back trajectories for all seasons were incorrectly shown as those for autumn. This error has now been corrected (Fig. R7 in this review comments reply letter), where the revised figure properly displays season-specific back trajectories. This correction does not affect any of the original data, analytical results, or scientific conclusions of the study. The key findings presented in Section 3.3 ("Impact of regional transport on aerosol hygroscopicity") remain fully valid and supported.

In addition, regarding the conclusion in lines 388–390, it is primarily speculative, suggesting that the observed more internal mixing may be caused by enhanced photochemical aging during summer (Mahish and Collins, 2017; Ray et al., 2023). We have made the following revisions in the manuscript: "This seasonal anomaly may be attributed to enhanced photochemical aging during summer transport… "

[Figure]

Figure R7. 72-hour air mass backward trajectories at a height of 100 meters corresponding to the cluster analysis during different seasons. The line colors denote different clusters, i.e., red for Cluster C1, green for Cluster C2, and blue for Cluster C3.

**Comment [16]:** "…lower hygroscopicity than in other areas…" Which areas is it compared to?

**Response:** We have modified the sentence (Line 409) to:" Aerosols in this suburban region exhibit relatively low hygroscopicity, which may be mainly due to relatively higher organic content (annual average $MF_{OA}$ in $PM_{2.5}$: 42.92%)."

**Comment [17]:** Which meteorological conditions are you referring to in line 436? Temperature, wind speed, wind direction, and RH for example are not mentioned in this manuscript. Local emission sources, besides traffic emissions were not linked to aerosol properties (lines 439).

**Response:** In the manuscript, we note that the discussion of meteorological conditions is limited to speculative explanations for certain phenomena. For instance, lines 287-289 state: "This behavior is mechanistically explained through temperature-mediated phase partitioning theory, where lower

nocturnal temperatures coupled with higher relative humidity facilitate efficient gas-to-particle conversion through aqueous-phase oxidation processes (Sun et al., 2013)." However, no substantive supporting data are presented, which may undermine the assertion that "These findings provide valuable insights into the complex interactions between aerosol chemical composition, particle size, seasonal meteorological conditions, and regional air mass transport in shaping aerosol hygroscopicity." Therefore, we have made the following revisions for clarity in the manuscript: " These findings provide valuable insights into the complex interactions between aerosol chemical composition, particle size, seasonal and regional air mass transport in shaping aerosol hygroscopicity. " The term "local emission sources" primarily refers to the impact of traffic emissions in the manuscript. Therefore, as stated in lines 439, it can be concluded that the study highlights the critical role of local emissions in influencing aerosol properties.

**Other comments:**

**Comment [18]:** Other comments Line 30: …suspended in a gas… is the commonly used definition. The HTDMA and CCNC measure in different saturation regimes. Thus, comparison of hygroscopicity measurements in the HTDMA and the CCNC is meaningless (lines 43-50). Please modify.

**Response:** We thank the reviewer for pointing this out. We have made the following revisions in the manuscript: " Aerosols, defined as mixtures of solid and liquid particles suspended in the air…". Furthermore, we have removed the sentence (lines 43-50) in the manuscript: " While these instruments can measure aerosol hygroscopicity, their measurement principles differ, and as a result, the resulting hygroscopicity data may show discrepancies (Liu et al., 2021; Liu et al., 2022; Ray et al., 2023; Zhang et al., 2017)."

**Comment [19]:** It would be nice to mention the ACSM in the introduction as it is used throughout the entire manuscript. Some statements are vague or confusing, for example lines 32-33, lines 40-41, lines 104 105, lines 223-224, lines 241-242. Line 212: 40 nm particles during winter and summer also show similar hygroscopicity in Fig. 1a Line 230: "…possesses greater hygroscopicity." compared to what? Line 231: "…more pronounced seasonal contrast." compared to what? One should be careful with words

like significantly or substantial (for example lines 231, 261, and 289). Please check throughout the manuscript.

**Response:** We have made the following series of revisions in the manuscript:

To avoid disrupting the manuscript's structure, the following statement has been added only at lines 98-99 of the introduction: " The H-TDMA observations enable the determination of size-resolved and seasonal variations in aerosol hygroscopicity in the Nanjing region. Furthermore, combining these data with aerosol chemical composition measurements facilitate the further analysis of influencing factors contributing to these hygroscopicity differences." Additionally, since the chemical composition data were jointly provided by ACSM and AE-33, the phrase " aerosol chemical composition measurements" is used here to refer to these two instruments.

Lines 32-33: "The hygroscopicity of aerosols describes their capacity to absorb water vapor, significantly influencing both the atmospheric environment and global climate through complex physicochemical processes. (Chen et al., 2019; Zhang et al., 2023). "

Lines 40-41: "However, aerosol hygroscopicity is influenced by various factors. Variations in environmental conditions and physicochemical processes can result in divergent hygroscopic behaviors across different atmospheric environments. "

Lines 104-105: This primarily illustrates the purpose of conducting this field observation experiment. Therefore, no modifications have been applied to the manuscript **in this section**.

Lines 223-224: "Furthermore, $\kappa_{LH}$ for 40–150 nm particles is slightly higher in spring compared to other seasons, while for 200 nm particles, it reaches slightly higher values in summer. Despite these variations, the variation in $\kappa_{LH}$ remains minor, typically ranging between 0.14–0.17, which is in the $\kappa$ range of SOA…"

Lines 241-242: "As demonstrated in Fig. 2b, the $CV_{\kappa\text{-PDF}}$ exhibits a decreasing trend with increasing particle size within the same season, reflecting enhanced internal mixing during aerosol aging from the Aitken mode to the accumulation mode…"

Line 212: " Furthermore, aerosols in winter and spring demonstrate enhanced hygroscopicity in the 80–200 nm size range compared to summer and autumn, which can be attributed to relatively higher content of SNA…"

Lines 230-231: "In general, aerosol particles exhibit higher $\kappa_{MH}$ in winter than in spring, which may be attributed to the higher abundance of $NO_3^-$ in winter aerosols leading to enhanced hygroscopicity of MH groups. This effect is more pronounced in accumulation-mode particles compared to Aitken-mode particles, with $\kappa_{MH}$ showing more distinct seasonal variations in the former…"

In addition, based on our review of the manuscript, we have made efforts to avoid using words such as "significantly" or "substantial". For example, line 384: "significantly higher $\kappa_{mean}$ values for 40 nm particles" has been revised to "relatively higher $\kappa_{mean}$ values for 40 nm particles", line 381-382: " despite substantial $NF_{NH}$ differences across air masses " has been revised to " despite relatively obvious $NF_{NH}$ differences across air masses "

**References:**

Aklilu, Y., Mozurkewich, M., Prenni, A.J., Kreidenweis, S.M., Alfarra, M.R., Allan, J.D., Anlauf, K., Brook, J., Leaitch, W.R., Sharma, S., Boudries, H., and Worsnop, D.R.: Hygroscopicity of particles at two rural, urban influenced sites during Pacific 2001: Comparison with estimates of water uptake from particle composition, Atmos Environ, 40, 2650-2661, https://doi.org/10.1016/j.atmosenv.2005.11.063, 2006.

Chen, J., Li, Z., Lv, M., Wang, Y., Wang, W., Zhang, Y., Wang, H., Yan, X., Sun, Y., and Cribb, M.: Aerosol hygroscopic growth, contributing factors, and impact on haze events in a severely polluted region in northern China, Atmos Chem Phys, 19, 1327-1342, 10.5194/acp-19-1327-2019, 2019.

Gysel, M., Crosier, J., Topping, D.O., Whitehead, J.D., Bower, K.N., Cubison, M.J., Williams, P.I., Flynn, M.J., McFiggans, G.B., and Coe, H.: Closure study between chemical composition and hygroscopic growth of aerosol particles during TORCH2, Atmos Chem Phys, 7, 6131-6144, 10.5194/acp-7-6131-2007, 2007.

Li, X., Chen, Y., Li, Y., Cai, R., Li, Y., Deng, C., Wu, J., Yan, C., Cheng, H., Liu, Y., Kulmala, M., Hao, J., Smith, J.N., and Jiang, J.: Seasonal variations in composition and sources of  atmospheric ultrafine particles in urban Beijing  based on near-continuous measurements, Atmos Chem Phys, 23, 14801-14812, 10.5194/acp-23-14801-2023, 2023.

Li, X., Chen, Y., Li, Y., Cai, R., Li, Y., Deng, C., Yan, C., Cheng, H., Liu, Y., Kulmala, M., Hao, J., Smith, J.N., and Jiang, J.: Seasonal variations in composition and sources of atmospheric ultrafine particles in urban Beijing based on near-continuous measurements, EGUsphere, 2023, 1-20, 10.5194/egusphere-2023-809, 2023.

Liu, Y., Li, H., Cui, S., Nie, D., Chen, Y., and Ge, X.: Chemical Characteristics and Sources of Water-Soluble Organic Nitrogen Species in PM2.5 in Nanjing, China, Atmosphere, 12, 574, 2021.

Mahish, M., and Collins, D.: Analysis of a Multi-Year Record of Size-Resolved Hygroscopicity Measurements from a Rural Site in the U.S., Aerosol Air Qual Res, 17, 1489-1500, 10.4209/aaqr.2016.10.0443, 2017.

Ray, A., Pandithurai, G., Mukherjee, S., Kumar, V.A., Hazra, A., Patil, R.D., and Waghmare, V.: Seasonal variability in size-resolved hygroscopicity of sub-micron aerosols over the Western Ghats, India: Closure and parameterization, Sci Total Environ, 869, 161753, https://doi.org/10.1016/j.scitotenv.2023.161753, 2023.

Shang, D., Hu, M., Tang, L., Fang, X., Chen, S., Zeng, L., Guo, S., Zhang, Y., and Wu, Z.: New Particle Formation Occurrence in the Urban Atmosphere of Beijing During 2013–2020, Journal of Geophysical Research: Atmospheres, 128, e2022JD038334, https://doi.org/10.1029/2022JD038334, 2023.

Shen, X., Sun, J., Che, H., Zhang, Y., Zhou, C., Gui, K., Xu, W., Liu, Q., Zhong, J., Xia, C., Hu, X., Zhang, S., Wang, J., Liu, S., Lu, J., Yu, A., and Zhang, X.: Characterization of dust-related new particle formation events based on long-term measurement in the North China Plain, Atmos Chem Phys, 23, 8241-8257, 10.5194/acp-23-8241-2023, 2023.

Sun, Y., Wang, Z., Fu, P., Jiang, Q., Yang, T., Li, J., and Ge, X.: The impact of relative humidity on aerosol composition and evolution processes during wintertime in Beijing, China, Atmos. Environ., 77, 927-934, https://doi.org/10.1016/j.atmosenv.2013.06.019, 2013.

Wang, J., Ge, X., Sonya, C., Ye, J., Lei, Y., Chen, M., and Zhang, Q.: Influence of regional emission controls on the chemical composition, sources, and size distributions of submicron aerosols: Insights from the 2014 Nanjing Youth Olympic Games, Sci Total Environ, 807, 150869, https://doi.org/10.1016/j.scitotenv.2021.150869, 2022.

Wang, X., Ye, X., Chen, J., Wang, X., Yang, X., Fu, T.M., Zhu, L., and Liu, C.: Direct links between hygroscopicity and mixing state of ambient aerosols: estimating particle hygroscopicity from their single-particle mass spectra, Atmos Chem Phys, 20, 6273-6290, 10.5194/acp-20-6273-2020, 2020.

Wang, Y., Li, Z., Zhang, R., Jin, X., Xu, W., Fan, X., Wu, H., Zhang, F., Sun, Y., Wang, Q., Cribb, M., and Hu, D.: Distinct Ultrafine- and Accumulation-Mode Particle Properties in Clean and Polluted Urban Environments, Geophys Res Lett, 46, 10918-10925, https://doi.org/10.1029/2019GL084047, 2019.

Xian, J., Cui, S., Chen, X., Wang, J., Xiong, Y., Gu, C., Wang, Y., Zhang, Y., Li, H., Wang, J., and Ge, X.: Online chemical characterization of atmospheric fine secondary aerosols and organic nitrates in summer Nanjing, China, Atmos Res, 290, 106783, https://doi.org/10.1016/j.atmosres.2023.106783, 2023.

Yuan, L., Zhang, X., Feng, M., Liu, X., Che, Y., Xu, H., Schaefer, K., Wang, S., and Zhou, Y.: Size-resolved hygroscopic behaviour and mixing state of submicron aerosols in a megacity of the Sichuan Basin during pollution and fireworks episodes, Atmos Environ, 226, 117393, https://doi.org/10.1016/j.atmosenv.2020.117393, 2020.

Zhang, S., Shen, X., Sun, J., Che, H., Zhang, Y., Liu, Q., Xia, C., Hu, X., Zhong, J., Wang, J., Liu, S., Lu, J., Yu, A., and Zhang, X.: Seasonal variation of particle hygroscopicity and its impact on cloud-condensation nucleus activation in the Beijing urban area, Atmos Environ, 302, 119728,

https://doi.org/10.1016/j.atmosenv.2023.119728, 2023.